# One-dimensional semimetal contacts to two-dimensional semiconductors

Xuanzhang Li [1], Yang Wei [1] ✉, Zhijie Wang [2], Ya Kong[3], Yipeng Su[1], Gaotian Lu[1], Zhen Mei[1], Yi Su[1], Guangqi Zhang[1], Jianhua Xiao[1], Liang Liang[1], Jia Li [2], Qunqing Li [1], Jin Zhang [3], Shoushan Fan [1] & Yuegang Zhang [1] ✉

Two-dimensional (2D) semiconductors are promising in channel length scaling of field-effect transistors (FETs) due to their excellent gate electrostatics. However, scaling of their contact length still remains a significant challenge because of the sharply raised contact resistance and the deteriorated metal conductivity at nanoscale. Here, we construct a 1D semimetal-2D semiconductor contact by employing single-walled carbon nanotube electrodes, which can push the contact length into the sub-2 nm region. Such 1D–2D heterostructures exhibit smaller van der Waals gaps than the 2D–2D ones, while the Schottky barrier height can be effectively tuned via gate potential to achieve Ohmic contact. We propose a longitudinal transmission line model for analyzing the potential and current distribution of devices in short contact limit, and use it to extract the 1D–2D contact resistivity which is as low as $10^{-6}$ $\Omega \cdot cm^2$ for the ultra-short contacts. We further demonstrate that the semimetal nanotubes with gate-tunable work function could form good contacts to various 2D semiconductors including $MoS_2$, $WS_2$ and $WSe_2$. The study on 1D semimetal contact provides a basis for further miniaturization of nanoelectronics in the future.

The extreme scaling of transistors is the relentless pursuit of integrated circuit industry[1,2]. Scaling down both channel length and contact length is necessary for the reduction of the overall device size. Two-dimensional (2D) semiconductors have shown great potential in the channel length scaling while reducing short-channel effects due to their atomically thin structures[3–5]. Field effect transistors (FETs) with sub-5 nm gate lengths can thus be developed with these materials[6–9]. Furthermore, some new strategies have also been proposed to optimize the contacts on the 2D channels, including van der Waals (vdW) contacts[10–12], edge contacts[13–17], introduction of thin tunnel barrier[18], ultra-high vacuum metal deposition[19] as well as semimetal bismuth contacts[20]. These efforts can effectively reduce the interface contact resistivity. However, it is still a great challenge to achieve ultra-short contact length with low contact resistance for high-performance 2D FETs. In addition to the contact resistance increase caused by

shortening of contact length (transmission line model)[21–23], the electrical conductivity of electrodes decreases seriously when the width is tailored to less than a critical value (typically ~10 nm). In the case of traditional 3D metal, the reduction of metal width leads to the reduced grain size and the enhanced electron scattering at grain boundaries. This results in a significant increase in resistivity[24–27] and even electromigration phenomena[28,29]. For 2D semimetals such as graphene, the length scaling will lead to quantization. Band gaps could be opened when the width of nanoribbon goes down to 10 nm[30,31]. Even if metallic nanoribbon can be realized by chirality design[32], the reduced carrier mobility caused by edge scattering will also limit its electrical performance[33,34]. Therefore, it is of great significance to explore new solutions to reduce the contact length of 2D transistors.

Metallic single-walled carbon nanotubes (SWCNTs) are the best-performance nano-scale quantum wires thanks to their perfect

[1]State Key Laboratory of Low-Dimensional Quantum Physics, Department of Physics and Tsinghua-Foxconn Nanotechnology Research Center, Tsinghua University, Beijing 100084, China. [2]Shenzhen International Graduate School, Tsinghua University, Shenzhen 518055, China. [3]College of Chemistry and Molecular Engineering, Peking University, Beijing 100871, China. ✉e-mail: weiyang@tsinghua.edu.cn; yuegang.zhang@tsinghua.edu.cn

quasi-1D single crystal structure[35], long distance ballistic transport behavior[36,37] and high current carrying capacity ($10^9$ A/cm²)[38]. Recent progresses have shown that they can be assembled with 2D materials to form mixed-dimensional vdW heterostructures with multiple functions[39–42]. Furthermore, the density of states (DOS) of the 1D semimetal SWCNT is small and almost constant between the two first van Hove singularities[43,44]. In such semimetal-semiconductor junctions, the clean and intact vdW interface as well as the suppressed metal-induced gap states (MIGS) in semiconductors can effectively eliminate the Fermi level pinning. The small DOS could also enable an efficient modulation on work function of CNTs by external electric field. Therefore, the gate-tunable Schottky barrier height (SBH) at such 1D/2D interface can be predicted. All these specific properties indicate that the individual semimetal SWCNT has great potential as an ultimate scaled contact in 2D electronics.

In this work, we propose a 1D semimetal contact by integrating two SWCNT electrodes with the same chirality on 2D semiconductors, which successfully reduces the contact length of 2D FETs to sub-2 nm. We demonstrate that such mixed-dimensional semimetal-semiconductor (Sm-S) junctions are more tightly contacted than the 2D–2D ones and can be switched between Schottky and Ohmic contact by tuning gate potential. A longitudinal transmission line model (LTLM)

was developed to describe the potential and current distribution of devices in short contact limit. The 1D-2D interfacial contact resistivity, as well as the contact resistance, were measured as $10^{-6}$ Ω·cm² and 50 kΩ·μm in Ohmic contact mode, which leads the way in ultra-short contacts to 2D semiconductors. Furthermore, the $MoS_2$, $WS_2$ and $WSe_2$ FETs with 1D semimetal contacts all exhibited high on/off ratio exceeding $10^6$ and on-state current of several microampere, indicating the generality of SWCNT contacts. This work indicates that FETs composed of 1D semimetal contacts and 2D semiconductors are prospective for future integrated circuits in the post Moore's law era.

## Results

### 1D semimetal contact

Figure 1a, b sketch the fabrication process of the 2D FETs with 1D semimetal contacts. The ultralong CNTs (centimeters in length) were grown with iron as catalyst and ethylene as carbon source by chemical vapor deposition (CVD)[45,46] (Fig. 1a). Suspended CNTs were obtained across the trenches on patterned silicon wafer (Fig. 1b-i). Evaporated sulfur particles were deposited on the surface of CNTs to assist the optical imaging[47] and precise manipulation of CNTs under optical microscope, as shown in Fig. 1b-ii and Supplementary Fig. 1. The entire individual ultralong CNT on Si substrate can be efficiently tracked with

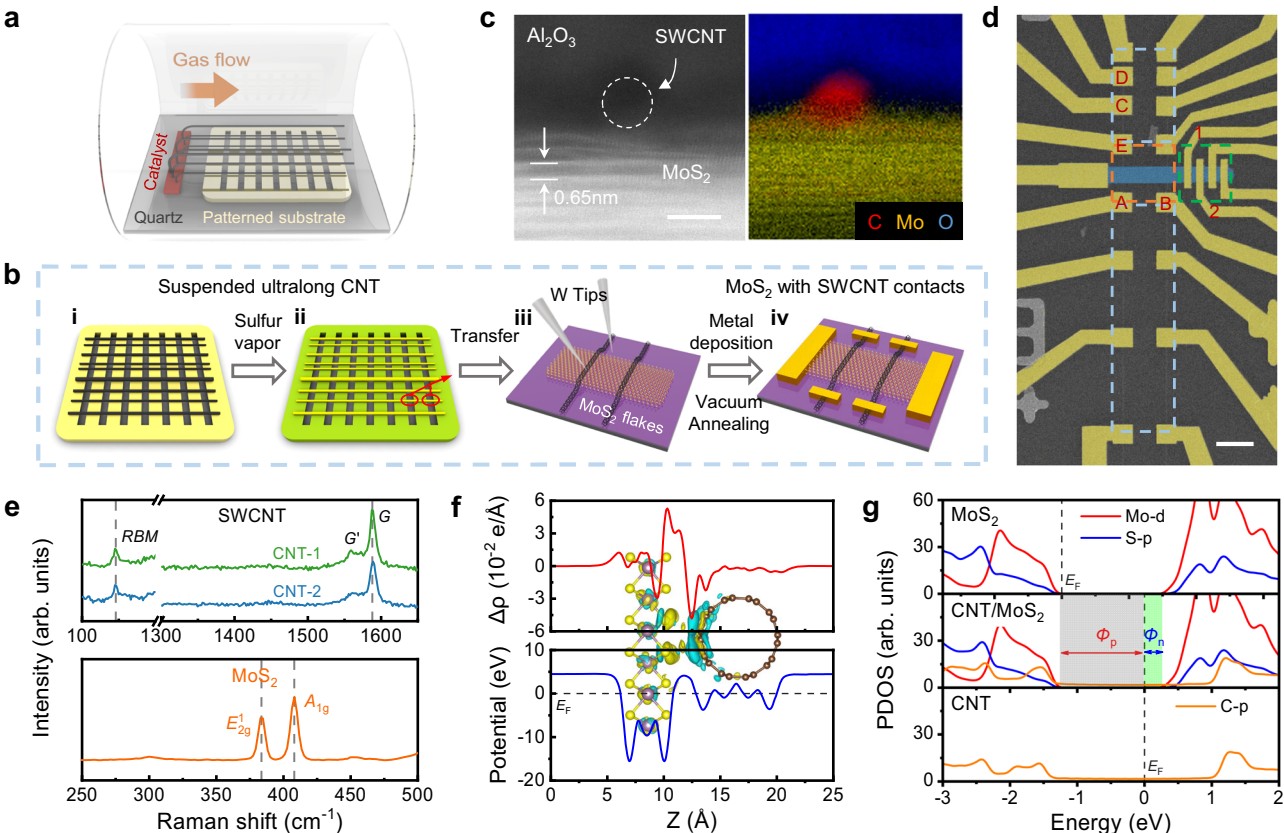

**Fig. 1 | Illustration and structural characterizations of 1D semimetal contact.** **a** Chemical vapor deposition of the ultralong CNTs. **b** Fabrication processes of the 2D FET with 1D semimetal contact: (i) sulfur treatment of CNTs; (ii) optically identification of the CNT position on the entire Si substrate; (iii) transfer two segments of a same metallic SWCNT (red circles in panel ii) onto the prepared rectangular TMD channel; (iv) vacuum annealing and metal deposition. **c** Cross-sectional STEM image and EELS map of a representative sample showing the $MoS_2$ channel and CNT electrode. Scale bar: 2 nm. **d** False-colored SEM image of the back-gate $MoS_2$ FET showing the $MoS_2$ channel (blue) and SWCNT electrodes (white). The dotted boxes present $MoS_2$ FET with 1D semimetal contacts (orange), CNT FETs with different channel lengths for transfer length method analysis (blue), and $MoS_2$ FETs for four-probe method measurements (green). Electrical properties of the

devices defined by electrodes A, B, C, D, E, 1 and 2 are shown in Fig. 2. Scale bar: 8 μm. **e** Raman spectra of individual CNT electrodes and the $MoS_2$ channel on Si/SiO₂ substrate using a 532 nm laser. The vertical dashed lines are used to mark the Raman peaks. The radial breathing mode (*RBM*) peaks of the SWCNTs verify their single-walled structure and same chirality. **f** Plane-averaged charge density difference and electrostatic potential of the (5,5) CNT/$MoS_2$ heterostructure along the Z direction (pointing from $MoS_2$ to CNT). The inset is the charge density difference of heterostructure with an isosurface is $7 \times 10^{-4}$ e/Å³, where yellow and cyan correspond to charge accumulation and depletion, respectively. $E_F$ is the Fermi level. **g** Projected density of states (PDOS) of $MoS_2$, (5,5) CNT before contact formation, and the 1D-2D heterojunction. The $\Phi_n$ and $\Phi_p$ represent the Schottky barrier for electrons and holes respectively.

the optical microscope (Supplementary Movie 1), and the metallic ones can be screened out in combination with resistance monitoring. Two suspending segments from a same metallic SWCNT across two neighboring trenches were picked by the W tips and then transferred onto a mechanically exfoliated 2D MoS$_2$ in parallel as the source and drain electrodes, respectively (Fig. 1b-iii). The specimen was then annealed at 300 °C under 400 mTorr Ar atmosphere for 1 h to remove the sulfur particles while achieving clean and tight mixed-dimensional vdW interface. More comparative experiments have further verified that the sulfur-assisted transfer technique does not affect the transport properties of both SWCNT and MoS$_2$, which can be found in Supplementary Note 1 and Supplementary Figs. 5, 6. Finally, the patterned Ti/Au (5 nm/50 nm) electrodes were fabricated by using e-beam lithography, electron beam evaporation and lift off procedures (Fig. 1b-iv).

Further characterizations were performed on the as-fabricated device. The 1D–2D interface between the SWCNT and MoS$_2$ was confirmed by a cross-sectional scanning transmission electron microscopy (STEM) image and the corresponding element spatial distribution captured by the electron energy-loss spectroscopy (EELS), as shown in Fig. 1c. Figure 1d is a scanning electron microscope (SEM) image of such a device, in which the two parallel and straight bright lines are the metallic SWCNT electrodes. The two 1D contacts defined a rectangular channel for the MoS$_2$ FET (orange dotted box). The electrodes in the blue and green dotted boxes were fabricated for further analysis on the interface contact resistivity. Figure 1e displays the Raman spectra of MoS$_2$ and the two SWCNTs on Si/SiO$_2$ substrate. More information about the Raman experiments can be found in Supplementary Fig. 4. The CNTs exhibit the prominent radial breathing mode (RBM) peaks around 146 cm$^{-1}$, verifying their single-walled structure and their same chirality. On Si/SiO$_2$ substrate, the quantitative relationship between the RBM peak and SWCNT diameter can be expressed as $\omega_{RBM} = 248/d_{CNT}$[48], and the diameter of CNT can be calculated to be 1.7 nm. The absence of $D$ peak around 1350 cm$^{-1}$ indicates the high quality of the CNT structure.

To present the geometry and electronic structures of the mixed-dimensional heterojunction, density functional theory (DFT) calculations were performed on a (5,5) CNT/MoS$_2$ heterostructure. The junction atomic structure, differential charge density and electrostatic potential distribution are presented in Fig. 1f. Comparative calculations were also conducted on a graphene/MoS$_2$ heterojunction (Supplementary Fig. 7). As shown by the variation of the relative energy as a function of interlayer distance, the most stable structure shows a vdW gap of 3.06 Å for CNT/MoS$_2$ and 3.42 Å for graphene/MoS$_2$, respectively. The obviously small vdW gap of such 1D–2D junction can be attributed to CNT's tubular structure. Since the tunneling probability of electrons decreases exponentially with the increase of barrier width, the smaller vdW gap indicates a smaller tunneling resistance. Figure 1g shows projected density of states (PDOS) of the heterostructure. The small and nearly constant DOS endows the CNT with gate-tunable work function, and efficiently suppresses the MIGS in MoS$_2$. Therefore, the gate-tunable SBH at the CNT/MoS$_2$ interface can be predicted. Without external electrical field, the Fermi level of CNT is 0.35 eV lower than the conduction band (CB) edge of MoS$_2$, indicating an $n$-type Schottky barrier. While with applied electrical field of 0.2 V/Å, the Ohmic contact can be achieved (Supplementary Fig. 9), the corresponding width $w_t$ and height $\Phi_t$ of the potential barrier are calculated to be 1.87 Å and 4.03 eV, respectively (Supplementary Fig. 10). This contributes to a small tunnelling resistivity of $2.882 \times 10^{-9}$ $\Omega \cdot cm^2$ (see details in Supplementary Note 7). These results present the potential of 1D semimetal contact for 2D FETs.

## Electrical transport measurements

Electrical transport measurements indicate that the ultra-short contact length of such 1D semimetal contacts can achieve improved performances of 2D FETs. Comparative transistors were made on the same MoS$_2$ flake with CNT contacts (smaller than 2 nm in diameter, electrodes A and B) and Ti/Au contacts (2 μm in contact length, electrodes 1 and 2), respectively. Transfer characteristics (black curve in Fig. 2a) of the individual SWCNT (electrodes C and D) present a conduction current about 5 μA and on/off ratio less than 5 under a bias voltage of 0.1 V, revealing good metallicity. The four-layered (4 L) MoS$_2$ transistor with CNT contacts exhibits n-type conduction with an on/off ratio exceeding 10$^7$ and on-state current of 4 μA, both of which are one order of magnitude larger than those of the device using Ti contacts. Supplementary Fig. 19 also shows the better switching performance of CNT contacts by comparing the devices using CNT, Ni and Au contacts. The linear and symmetrical output characteristics (Fig. 2b) indicate that Ohmic contact with negligible Schottky barrier can be acquired at the CNT-MoS$_2$ interface under positive $V_g$. In contrast, the I–V curves of the device using Ti contacts show obvious nonlinearity (Fig. 2c), implying the presence of barriers. This can be further verified by temperature-dependent transfer characteristics. As temperature lowering, the on-current of CNT-MoS$_2$ FET increases due to the enhanced mobility of MoS$_2$, whereas the current of Ti- MoS$_2$ FET decreases for the weakened thermal emission current (Fig. 2d). Furthermore, the barrier heights $\Phi_B$ at the CNT-MoS$_2$ and Ti-MoS$_2$ interfaces have been extracted as a function of $V_g$ from the slops of $\ln(I_{ds}/T^{3/2}) \sim q/kT$ (Fig. 2e) following the 2D thermal emission theory[10], and the results are presented in Fig. 2f. It can be seen that the $\Phi_B$ at CNT-MoS$_2$ interface changes significantly in the lower $V_g$ regime and the tuning efficiency of the barrier decreases at higher $V_g$. In comparison, the barrier height at CNT-MoS$_2$ interface can be modulated more efficiently than that at Ti-MoS$_2$ interface. Ohmic contact can be achieved for the CNT contact at $V_g = 35$ V, where the barrier height drops to 0. The FET composed of monolayer (1L) MoS$_2$ behaves similarly, which can be found in the Supplementary Fig. 13.

The comparative band diagrams in Fig. 2g further reveal the advantages of semimetal CNT contacts, which can be deduced from the CIFS analysis in the Supplementary Note 4 and Supplementary Fig. 15. For Ti/Au, the SBH at the metal-semiconductor (M-S) interface is practically determined by the Fermi level pinning effect and is gate-independent, which places great challenges on the realization of high-performance Ohmic contact. While for CNT, the SBH at the Sm-S interface can be efficiently modulated because of the absence of Fermi level pinning and the gate-tunable work function of CNT. The SBH can be reduced to zero and Ohmic contact can thus be achieved. In addition to SBH, the $\Phi_B$ modulation is also assisted by the tuning efficiency of MoS$_2$'s Fermi level, which is more efficient at lower $V_g$ than at higher $V_g$ (see details in Supplementary Note 5). Briefly, the gate modulation process of the FETs with CNT contacts can be depicted in Fig. 2g. At $V_g = -8$ V, the interface barrier height is above 600 meV and the device is at off state. As $V_g$ is tuned from negative to positive, the work function of CNT, as well as the SBH significantly decreases. At $V_g = 7.5$ V, the barrier height of the 1D contact is 191 meV, and the device is at the subthreshold region. When $V_g = 50$ V, the Fermi level of CNT enters the conduction band of MoS$_2$, indicating an Ohmic contact is achieved.

## LTLM and extraction of contact resistance

The contact resistance $R_c$ ($\Omega \cdot \mu m$) is a critical parameter for quantitative evaluation of the 1D semimetal contact. It is a combination of the interface contact resistivity $r_c$ ($\Omega \cdot cm^2$) and the channel sheet resistivity $\rho_{TMD}^{2D}$ ($\Omega \cdot \square^{-1}$) for 3D or 2D metal contacts according to the conventional transmission line model. In the case of the 1D semimetal contact, the current is efficiently injected into the channel (Supplementary Fig. 16e), and the contact resistance in short contact limit should be expressed as $R_c = r_c/l_c$ (see details in Supplementary Note 7). The

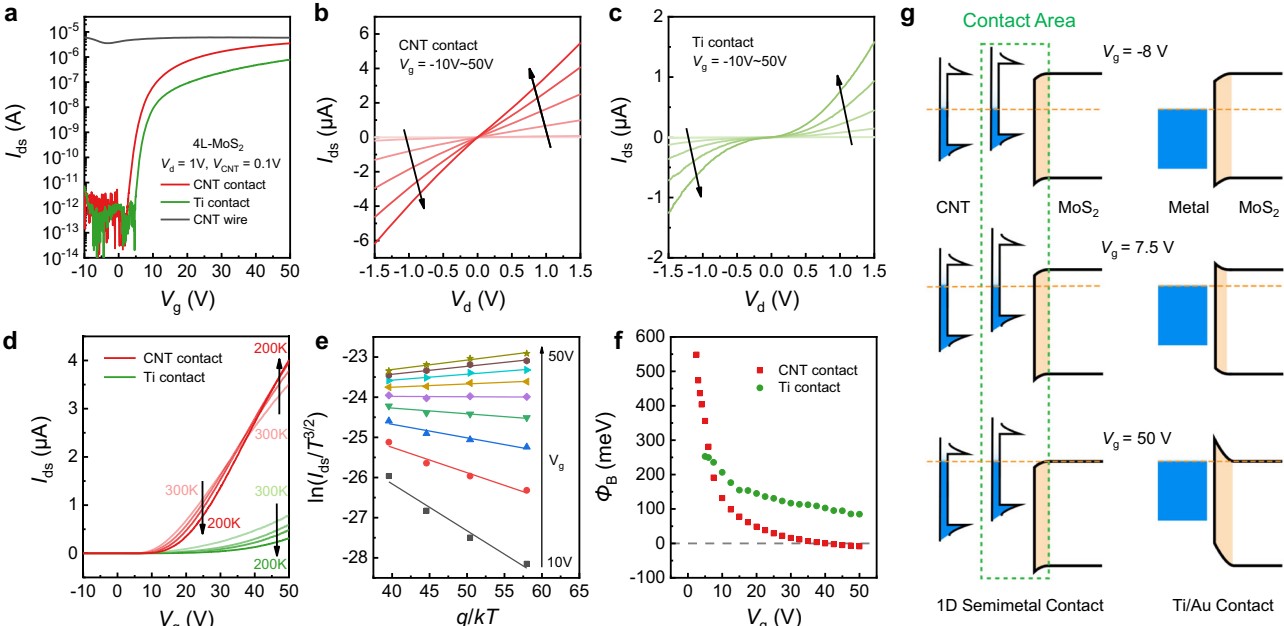

**Fig. 2 | Electrical transport measurements of MoS₂ FETs with 1D semimetal contacts. a** Room-temperature transfer characteristics of a 4L-MoS₂ FET with SWCNT (red) and Ti (green) contacts on 300 nm SiO₂ dielectrics. The $I_{ds}$ and $V_g$ are conduction current in channel and gate voltage respectively. **b** Output characteristics of the MoS₂ FET with SWCNT contacts. The $V_d$ is drain bias voltage. **c** Output characteristics of the MoS₂ FET with Ti contacts. **d** Temperature-dependent transfer characteristics of the MoS₂ FET with SWCNT (red) and Ti (green) contacts. **e** Arrhenius plots of the CNT-contacted four-layer MoS₂. The $q$, $k$ and $T$ are

elementary charge, Boltzmann constant and temperature respectively. **f** Gate-dependent barrier height of the CNT/MoS₂ (red) and Ti/MoS₂ (green) interfaces. **g** Comparison band diagram for 2D FETs with 1D semimetal contact and conventional metal contact. The orange dashed lines represent the Fermi levels. The barrier height at semimetal-semiconductor interface is gate-tunable due to the gate-dependent work function of semimetal, while the one at metal-semiconductor interface is nearly fixed.

contact length $l_c$ is defined as the diameter of CNT (the feature size of contact geometry).

Due to the micron-scale 2D FET device size (on the same order of magnitude as the mean free path of electrons in metallic SWCNT), the resistance of CNT electrodes is not negligible, and the potential/current distribution in the channel is not uniform. Thus, the commonly used four-probe method or transfer length method is not applicable for the measurement of the 1D contact resistance. Considering the symmetric conditions of our devices, an LTLM can be developed to describe the potential and current distribution of devices in short contact limit (details can be found in Supplementary Note 6). The 1D–2D heterostructure can be modeled as a resistor network, as sketched in Fig. 3a. The total resistance of the device $R_{tot}$ (two-probe resistance between Source and Drain) includes three parts in series: the contact resistance $R_1$ between CNT and Ti, the interconnect resistance $R_2$ of the CNT wires on SiO₂ substrate, and the resistance $R_3$ of the heterostructure region. They can be expressed as:

$$R_{tot} = R_1 + R_2 + R_3 \qquad (1)$$

where $R_1 = 2R_{nc}^{CNT} + R_Q^{CNT}$, $R_2 = 2\rho_{CNT}^{on\,SiO_2}L_{in}$, $R_3 = R_H + R_Q^{CNT}$. $R_Q^{CNT}$, $R_{nc}^{CNT}$, $\rho_{CNT}^{on\,SiO_2}$, $L_{in}$ is the quantum resistance of CNT, the interfacial resistance between CNT and Ti, the resistivity of the CNT wire, and the interconnect length of CNT wire, respectively. $R_H$ is the equivalent resistance of the heterostructure:

$$R_H = \sqrt{2R_t\rho_l}\big/\left(1 - e^{-\frac{W}{L_T}}\right) \qquad (2)$$

where $R_t$ (Ω·μm) is defined as the resistance experienced by the transverse component of current that is injected into MoS₂ channel and transport along the $x$ direction; $\rho_l$ (Ω·μm⁻¹) is defined as the resistance experienced by the longitudinal component of current that is transmitted in CNT wires along the $y$ direction; $L_T$ is a characteristic

length to evaluate the decay rate of current density along the $y$ direction. They can be expressed as:

$$R_t = 2r_c/D_{CNT} + \rho_{TMD}^{2D}L, \rho_l = \rho_{CNT}^{on\,MoS_2}, L_T = \sqrt{R_t/2\rho_l}$$

where $\rho_{TMD}^{2D}$, $\rho_{CNT}^{on\,MoS_2}$, $D_{CNT}$, $L$ and $W$ is the sheet resistivity of MoS₂, the resistivity of CNT electrodes in contact with MoS₂, the diameter of CNT, channel length, and channel width, respectively; $r_c$ is the 1D–2D interface contact resistivity, to be deduced from our model and measurement results.

The parameters $\rho_{CNT}^{on\,SiO_2}$, $\rho_{CNT}^{on\,MoS_2}$, $R_{nc}^{CNT}$ can be obtained by measuring the CNT FETs with different channel lengths (blue and orange dotted boxes in Fig. 1c)[49]. Since the current flowing through MoS₂ is three orders of magnitude smaller than that in CNT, the influence of MoS₂ can be neglected (electrodes A and E). The results were presented as the blue, green and black curves shown in Fig. 3b and inset of Fig. 3c, respectively. Meanwhile, the four-probe measurement of the Ti-MoS₂ FETs (green dotted box in Fig. 1c) can determine $\rho_{TMD}^{2D}$. Figure 3c plots $\rho_{TMD}^{2D}$ of 1 L and 4 L MoS₂ as a function of $V_g$. The $L_{in}$, $L$ and $W$ were measured by atomic force microscope (AFM), as shown in Figure S3a, b. The $D_{CNT}$ was determined to be 1.7 nm from the RBM peak position of CNT Raman spectrum.

On the basis of these measurements and Eq. 2, the 1D-2D interface contact resistivity ($r_c$) and the contact resistance ($R_c$) can be acquired. The $r_c$ between the CNT and the 1L and 4L MoS₂ is shown in Fig. 3d. The contact resistivity as low as $10^{-6}$ Ω·cm² can be achieved at high gate potentials for both 1L and 4L MoS₂. Under these gate voltages, the heterojunctions are in Ohmic contact mode and the contact resistivity is independent of their initial band alignment. The heterojunction composed of 1L MoS₂ exhibited larger contact resistivity at low gate bias due to its larger band gap and the higher Schottky barrier at the Sm-S interface. Figure 3e compares the contact resistance $R_c$ of this work with the state-of-the-art contacts in literatures including Ti,

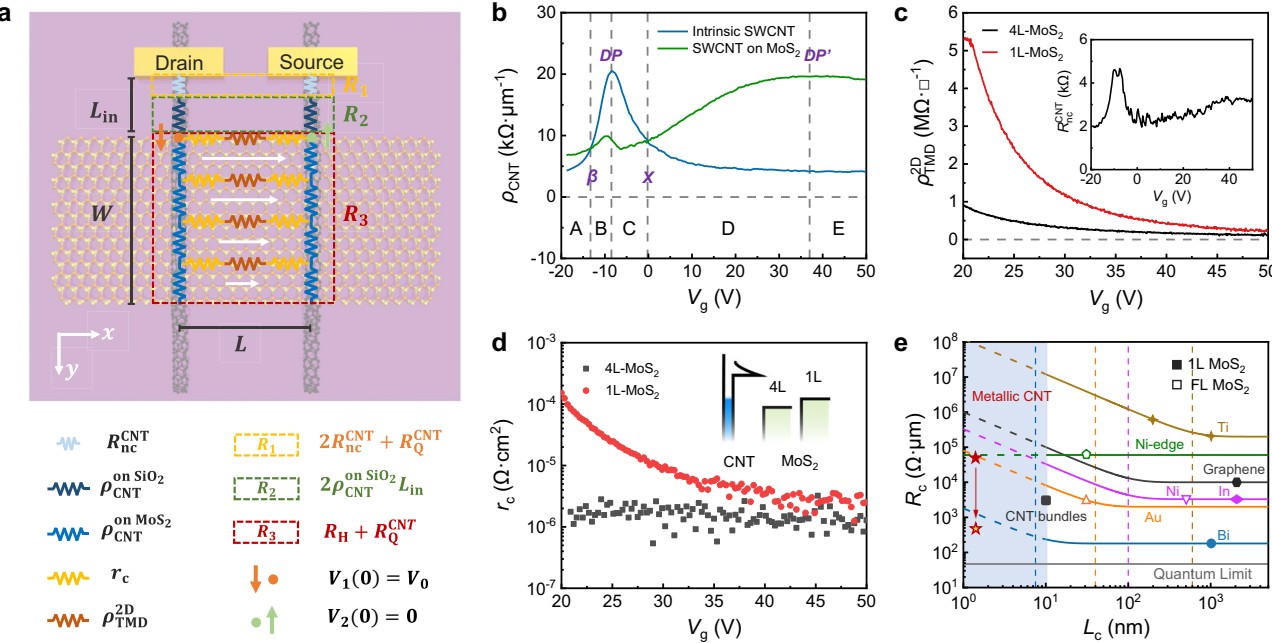

**Fig. 3 | Contact resistivity analysis and benchmark of 1D semimetal contact.**
**a** Top view and equivalent circuit of the MoS₂ FET with 1D semimetal contact. Dotted boxes present the contact resistance between CNT and Ti ($R_1$, yellow), the interconnect resistance of the CNT wire ($R_2$, green) and the overall resistance of the heterostructure ($R_3$, red). The white arrows represent the conduction current in channel. The $W$, $L$, and $L_{in}$ are channel width, channel length and interconnect length of CNT wire. The shown resistance/resistivity quantities include the quantum resistance of CNT ($R_Q^{CNT}$), the interfacial resistance between CNT and Ti ($R_{nc}^{CNT}$), the resistivity of CNT wire ($\rho_{CNT}^{on\,SiO_2}$), the resistivity of CNT electrodes in contact with MoS₂ ($\rho_{CNT}^{on\,MoS_2}$), the 1D–2D interface contact resistivity ($r_c$), the sheet resistivity of MoS₂ ($\rho_{TMD}^{2D}$) and the equivalent resistance of the 1D–2D heterostructure ($R_H$).
**b** Measured resistivity of the CNT with and without MoS₂ contacting. The feature points $\beta$, $X$, $DP$ and $DP'$ (marked by the vertical dashed lines) and the resulting five

regions A-E are used for the CIFS analysis in Supplementary Note 4. **c**, Measured sheet resistivity of 1L and 4L MoS₂. Inset is the interface resistance between CNT and Ti. **d** Extracted interface contact resistivity of the CNT/1L-MoS₂ and CNT/4L-MoS₂ heterojunction. Inset shows the band offset between different materials. **e** State-of-the-art contact technology for MoS₂ transistors plotted as a function of contact length. The gray line represents the quantum limit of $R_c$ with the carrier density $n_{2D} = 5 \times 10^{12}$ cm⁻². The polygons represent the data reported in experiments. The corresponding curves are theoretical deduction and the vertical dashed lines are transfer length for different contacts, they are all given by the Transmission Line Model (Supplementary Note 7). The blue shaded area indicates the area where the resistivity of conventional 3D/2D metal increases sharply due to the limitation of size effect. The solid red star represents the experimental measured data and the one with yellow core is the theoretical calculated result.

Ni, Au, In, Graphene, Bi and CNT bundles (refs. [11], [14], [19–21], [50–52]). The polygon symbols present the experimental results, and the curves represent the theoretically deduced $R_c$ versus contact length $l_c$ (given by the transmission line model, see details in Supplementary Note 7). The contact resistance increases rapidly with the reduction of contact length when $l_c < l_t$ ($l_t = \sqrt{r_c / \rho_{TMD}^{2D}}$). The blue shaded area ($l_c < 10$ nm) is difficult to reach using 3D/2D metal contacts, and therefore, no experimental result exists before this work. The introduction of 1D semimetal contact successfully reduces the contact length by an order of magnitude to sub-2 nm, while its corresponding contact resistance maintains a relatively small value about 50 kΩ·µm (the solid red star in Fig. 3e). It should be noted that the experimentally measured $r_c$ of the CNT contact is still about two orders of magnitude larger than the theoretical tunneling resistivity ($2.882 \times 10^{-9}$ Ω·cm²). This means that the 1D semimetal contact could potentially further be reduced to an ultra-low contact resistance of 419 Ω·µm (the hollow red star in Fig. 3e). Moreover, the field-effect mobilities of the 4L and 1L devices were calculated to be 85.9 and 69.3 cm² V⁻¹ s⁻¹, respectively, after taking into account the inhomogeneous distribution of currents (see details in Supplementary Note 9). Overall, in short contact limit, the 1D semimetal contact using individual SWCNT is prominent over most of the conventional metal contacts as well as the family of semimetal carbon nanomaterials including graphene and CNT bundles. Such performance can be attributed to the Ohmic contact achieved by gate modulation, the smaller vdW gap between 1D and 2D materials, and the perfect quasi-1D single crystal structure of CNT electrodes. All

these results and comparison indicate that the SWCNT contact is a promising solution for the ultimate scaling of 2D transistors.

## General semimetal CNT contact

Benefiting from the clean and intact vdW interface as well as the small DOS of CNT, the CNT electrodes perform well as ultra-short contacts for 2D MoS₂ FET. To further evaluate their capability as general electrodes for various 2D semiconductors, MoS₂, WS₂, and WSe₂ FETs with 1D semimetal contacts were fabricated on 2D h-BN flakes, where an Au bottom gate was set beneath the BN dielectric layers (inset of Fig. 4a). The solid curves in Fig. 4b present the transfer characteristics of the three FETs. They all exhibit on/off ratio exceeding 10⁶ and subthreshold swing lower than 250 mV/dec. The different threshold voltages come from their different band edge positions. The negative threshold voltage of this MoS₂ device compared with the one in Fig. 2a should be attributed to their different gate materials. The linear output characteristics of MoS₂ (Fig. 4c) and WS₂ (Fig. 4d) FETs under high gate voltage confirm the Ohmic contacts, while the WSe₂ FET shows excellent ambipolar conduction characteristics. The arrows with gradient color indicate the direction of $V_g$ sweeping. All the three FETs with CNT contacts exhibit on-state current of several microampere. By contrast, the comparative FETs using Ti/Au contact (dashed curves in Fig. 4b) show poor performance for all the three channel materials due to Fermi level pinning and fixed SBHs. The compatibility with various 2D materials could be further attribute to the efficient modulation on Fermi level of CNT. The work function of CNTs

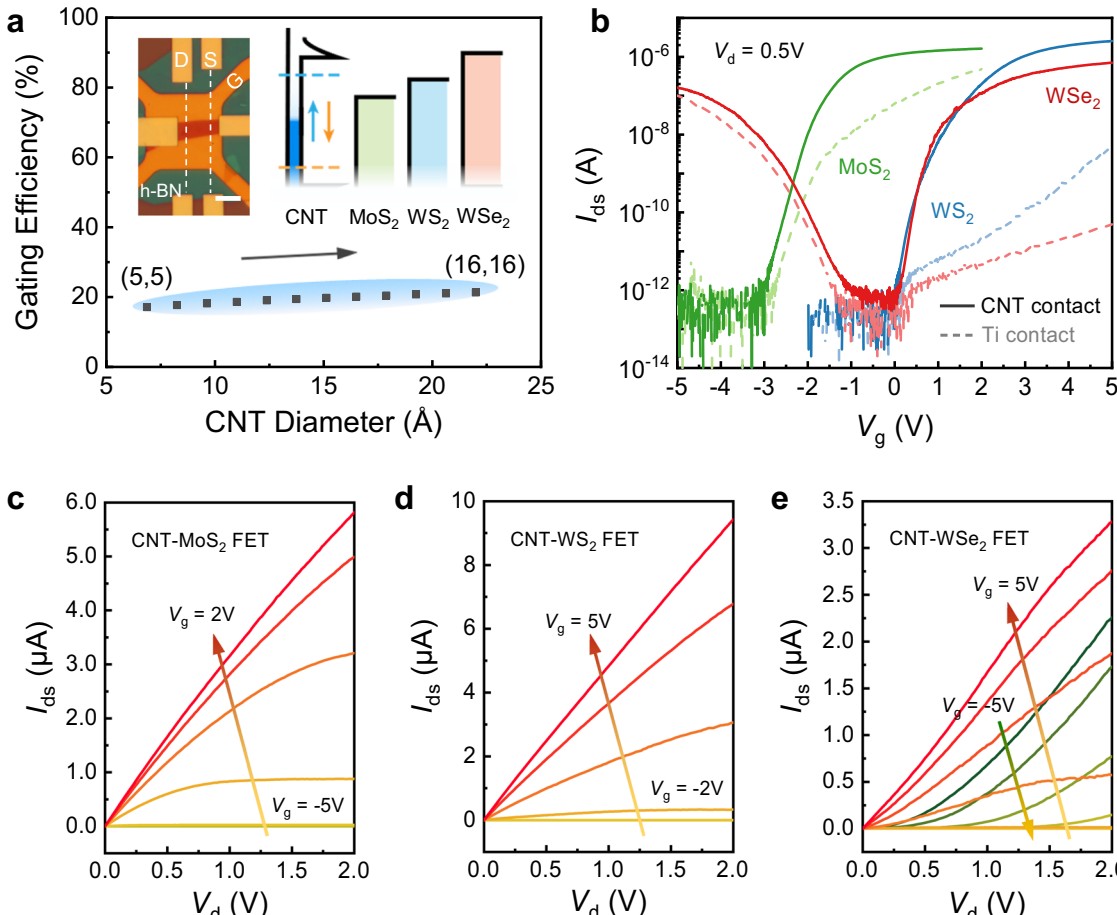

**Fig. 4 | General semimetal CNT contacts. a** Calculated gating efficiency ($d\mu_{CNT}/dV_g$) of armchair CNTs using 20 nm-thick h-BN dielectric. Inset shows the optical image of the as-fabricated device and the band offset between different materials. The white vertical dashed lines represent two CNT electrodes. Scale bar: 5 μm. The two horizontal dashed lines represent the gate-tunable work function of CNT. **b** Room-temperature transfer characteristics of local bottom-gate FETs with MoS$_2$, WS$_2$, and WSe$_2$ channels. Output characteristics of MoS$_2$ (**c**), WS$_2$ (**d**), and WSe$_2$ (**e**) FETs, showing $V_g$-dependent I–V relationship and on-state current of several microampere. The arrows with gradient color indicate the direction of $V_g$ sweeping. The $V_g$ step is 1 V.

can be tuned by $V_g$ through electrostatic capacitance effect following $\frac{1}{e}\frac{dW_{CNT}}{dV_g} = -\frac{1}{e}\frac{d\mu_{CNT}}{dV_g} = -\frac{C_g}{C_Q + C_g}$, where $C_g$ and $C_Q$ are the gate capacitance and quantum capacitance[53,54] respectively. Figure 4a presents the calculation result of several armchair SWCNTs on 20 nm-thick h-BN gate dielectric, and the calculated gating efficiency is around 20%. This indicates that the Fermi level can be tuned in a wide range, which can effectively match the band edge of various 2D materials and make sure good contacts (inset of Fig. 4a). Therefore, good contacts can be accessed for a variety of device design requirements including the threshold voltage and conduction type by employing semiconductors with different band edge positions. This further demonstrates the ability of the 1D semimetal contacts to be applied in high density and low power logic circuits.

## Discussion

In summary, we report the 1D semimetal contacts to 2D semi-conductors, which successfully reduce the contact length to sub-2 nm. The FETs with such 1D contacts exhibited enhanced performance. The potential and current distribution of devices in short contact limit was modeled by the as-proposed LTLM and the 1D−2D interface contact resistivity was thus measured as low as $10^{-6}$ Ω·cm². The low contact resistance can be attributed to the formation of Ohmic contact and the smaller vdW gap at the 1D-2D interface. The SWCNTs demonstrated their ability to form good contact with various semiconductors, which

strongly supports the miniaturization of low-power electronic devices and the extending of Moore's Law in the future.

## Methods
### Synthesis of ultralong CNT

The ultralong CNTs are grown using a CVD system. The catalyst strips are prepared by electron beam deposition of 0.2 nm Fe onto the silicon substrate and then slicing. The furnace temperature is set to 970 °C for CNT growth. The H$_2$ and C$_2$H$_4$ with flow rate of 200 sccm and 1 sccm are selected as reducing gas and carbon source, respectively. A pre-prepared Si/SiN$_x$ substrate with seven 200 μm-wide trenches is placed behind the catalyst strip to collect the CNTs grown in suspension. The as-grown centimeter-long CNTs with suspending sections over trenches can be observed in Supplementary Fig. 1 and Supplementary Movie 1.

### Selection of metallic SWCNT

The sulfur powder is heated to 150 °C via a hot plate for evaporation. The fresh ultralong CNTs placed on trenched Si/SiN$_x$ substrate are then treated in sulfur atmosphere for 10 s. Empirically, the condensed morphology of sulfur vapor on CNTs depends on the diameter of CNTs and the sparser distribution of sulfur droplets always indicates the smaller diameter. Select the freestanding SWCNTs with the sparsest sulfur droplet distribution under the optical microscope. W tips connected to Keithley 2900 are used for judging the CNT conductivity.

Since CNTs owns various helical structures, the $I-V$ measurements, Raman spectroscopy and AFM images are needed to fully confirm their single-walled structure and metallicity after the device is prepared.

## Measurements

Electrical measurements were carried out on a semiconductor analyzer (Agilent B1500A) combined with a Lakeshore CRX-4K under vacuum. Raman spectroscopy was performed on Horiba Jobin Yvon LabRAM HR 800 with a 532 nm focused laser. SEM images were obtained on a FEI Nova NanoSEM 450 with accelerating voltage at 1 kV. The cross-sectional STEM samples were prepared using a FEI Helios G4 PFIB CXe. The STEM images and EELS map were performed using a FEI Titan 80–300 with a double-sided spherical aberration corrector at 300 keV.

## Density functional theory calculations

All DFT calculations in this study were performed within the Vienna *Ab-initio* Simulation Package (VASP)[55]. The projector augmented wave (PAW) potentials[56] and generalized gradient approximation (GGA) with the Perdew-Burke-Ernzerhof (PBE) functional[57] were used to describe the electron-ion interaction and exchange-correlation energy, respectively. A representative (5, 5) SWCNT with the $5 \times 1 \times 1$ supercell and a monolayer $MoS_2$ with the $4 \times 3\sqrt{3} \times 1$ supercell were taken into account to establish the $CNT/MoS_2$ heterostructure. The DFT-D3 correction method[58] was used to describe the van der Waals interaction. The cutoff energy of the plane wave basis was set to 400 eV. The convergence criteria for the total energy and force were set to $10^{-5}$ eV and $0.02$ eV Å$^{-1}$, respectively. The vacuum distance of at least 12 Å in the $z$ direction was imposed to eliminate the interactions between the periodic images.

## Data availability

Relevant data supporting the key findings of this study are available within the article and the Supplementary Information file. All raw data generated during the current study are available from the corresponding authors upon request.

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

## Acknowledgements
This work was financially supported by the National Key Research and Development Program of China (Grant No. 2018YFA0208401 and 2021YFA1400100), the National Natural Science Foundation of China (Grant Nos. 61774090, U1832218, 51727805), the National Key Research and Development Program of China (Grant No. 2017YFA0205803), the Key-Area Research and Development Program of Guangdong Province (Grant No. 2020B010169001), the Local Innovative and Research Teams Project of Guangdong Pearl River Talents Program (2017BT01N111) and Basic Research Project of Shenzhen, China (JCYJ20200109142816479).

## Author contributions
Y.W. and Y.Z. conceived and supervised the research. Y.W. and X.L. designed the experiments. X.L. performed experiments on synthesis of CNTs, device fabrication, characterization, measurements, and data analysis with help from G.L., Z.M., G.Z., J.X., and L.L. Z.W. carried out the density functional theory calculations. Y.K. performed the Raman and AFM measurements. Y.P.S. participated in the modeling of LTLM. Y.S. conducted the data analysis of STEM and EELS. J.L., Q.L., J.Z., and S.F. participated in data analysis. The paper was written by X.L., Y.W., and Y.Z. with contributions from all the co-authors.

## Competing interests
The authors declare no competing interests.
