## [Peer Review File · Nature Communications]

One-Dimensional Semimetal Contacts to Two-Dimensional SemiconductorsReviewers' Comments:

Reviewer #1:

Remarks to the Author:

Review comments are attached.

In the manuscript entitled “One-Dimensional Semimetal Contacts to Two-Dimensional Semiconductor”, Li et al., report a novel junction contact between 1D-SWCNT and 2D-MoS₂. This work presents some interesting and well-analyzed results. However, it seems to have a few issues to be considered before publication.

1. In Figure 1a, how are the authors sure that the sulfur particles were removed after annealing? Since the deposited sulfur particles can affect the charge carrier concentration and electrical properties of the CNTs, it is necessary to show the evidence of the removal of the sulfur particles. It is easy to distinguish the doped CNTs and undoped CNTs by the Raman spectroscopy [S. Suzuki et. al. Carbon 49, 2264-2272 (2011)] and transfer characteristics [Ref. 41]. I suggest to measure the Raman spectra and transfer curves of the CNTs at each step of fabrication processes (Figure 1 a-ii-v).
2. As MoS₂ is very vulnerable to the temperature-dependent sulfur vacancy, the authors need to describe the desulfurization process of CNT in more details (including the environments like temperature, atmosphere, and time.). Also, to shoot out the defect issue of the MoS₂, the authors need to provide clear evidences about the state of MoS₂ via a comparative assessment between pre-/post-treated MoS₂ samples.
3. The RBM mode of CNT can span from $\sim 130\text{ cm}^{-1}$ to $\sim 300\text{ cm}^{-1}$, depending on its van Hove electronic transition, size, and chirality, etc. However, it is difficult to evaluate this due to the breaking markers. The authors need to roll back the break regime. Like the author’s statements, if the vdW gap between SWCNT and 2D MoS₂ is so small, it is not suitable to monitor the post-transferred CNT solely. The authors need to proceed the comparative Raman study about both the pre-transfer suspended CNT and the post-transfer CNT in order to evaluate how RBM mode is influenced by the vdW interaction between them. If there occurred some shift due to the interaction, the current radius result on the transferred state can be modified somewhat.
4. There is no comment on the linear line of low gate bias regime in Figure 2f and Figure 7f. Also, Figure S7(f) is not matched with the corresponding caption. Where is the green plot?
5. As stated in Ref. 10, the barrier height (Φ_B) can be generally elucidated by the linear combination of work function of CNT, the work function, and affinity of n-type MoS₂ (that is, $\Phi_B = W_{\text{CNT}} - W_{\text{MoS}_2} + \chi_{\text{MoS}_2}$). Assuming that W_{CNT} , W_{MoS_2} , χ_{MoS_2} in the ambient condition have the typical values of $\sim 4.5\text{ eV}$, $\sim 4.0\text{ eV}$, $\sim 5.1\text{ eV}$, respectively, the barrier height at $V_g = 0\text{ V}$ is predicted to be roughly as large as $\sim 3.4\text{ eV}$. However, Figure 2f shows that Φ_B is at most $\sim 0.6\text{ eV}$ at 0 V . What is the reason to make such a big difference?
6. In Figure 2f, the decay rate difference of Φ_B between the two curves seems to be very similar in the high bias regime ($V_g > 15\text{ V}$). On the other hand, its bias-induced reduction is pretty remarkable in the lower bias regime ($V_g < 10\text{ V}$). Thus, CNT-effect seems to play a more dominant role in lowering the barrier in the low bias regime than in the higher bias regime. It would be better to check if the CNT/MoS₂ sample follows the Square-root-type behavior of SWCNT work function as reported by Yu et al. (Nano Letters 9, 3430 (2009)).
7. As shown in the CNT-contacted sample of Figure 2d, the current in the V_g -region lower than 35 V decreases with decreasing temperature ($V_g < 35\text{ V} \rightarrow$ insulating), while it increases with temperature for the higher bias region ($V_g > 35\text{ V} \rightarrow$ metallic). The insulating and metallic properties are related to the barrier height in the corresponding voltage regimes of Figure 2f. That is, the insulating property lasted until the barrier height reaches zero. So does Ti-contacted case in Figure 2d. However, in Figure S7, although the barrier height is nearly 0 eV above 40 V , the current increases with temperature in the overall voltage regime.
8. In Figure S4, the authors need to explain why the vdW gap of 3.08 \AA comes out. Is the vdW gap value independent to the chirality of SWCNT? Can this small gap value be obtained only in the CNT with (5,5)?
9. It is required to compare the CNT contacts with graphene contacts. Although the interlayer distances between CNT/MoS₂ and graphene/MoS₂ are slightly different ($\sim 0.36\text{ \AA}$), Is there a significant difference in the performance of devices with CNT and graphene contacts?
10. The author described the longitudinal transfer length method (LTLM) method to estimate the contact resistance of between CNT/MoS₂ channel. The equivalent circuit model was well designed, and the most

of parameters used in calculation were also well measured, separately. However, I am concerned about the measurement of $\rho_{\text{CNT}}^{\text{on MoS}_2}$. The authors mentioned that the $\rho_{\text{CNT}}^{\text{on MoS}_2}$ was measured at the device in the orange box of Figure 1c. Since the CNTs are placed on the MoS₂, the measured current can be influenced by the MoS₂. How can the authors be sure that the measured current shows only the resistivity of CNTs?

11. The authors mentioned that the Fermi level pinning at the CNT/MoS₂ contact is suppressed by small DOS of the CNT. However, since CNTs were transferred onto MoS₂, the interface between the two materials form a van der Waals contact. Therefore, the suppression of pinning might be due to the van Waals gap between the MoS₂ and CNT (Nature 557, 696–700, (2018)).
12. In case of the 1D semimetal contact, author explained that injection efficiency of electrons from electrode to channel is greatly improved, and the contact resistance at the short contact limit shall be expressed as $R_C = r_C/D_{\text{CNT}}$. According to this expression, it is likely that CNT with a diameter larger than 2nm is better for improving the contact properties. Is there any experimental data using CNT with various diameter?

Reviewer #2:

Remarks to the Author:

The paper by X. Li and colleagues reports on the electrical characteristics of 2D semiconductor FETs, e.g., MoS₂, WSe₂, contacted directly with semi-metallic carbon nanotubes as source and drain electrodes. The authors describe an involved fabrication procedure, perform DFT calculations to estimate the electrostatic nature of the CNT/MoS₂ interface, and report the experimental transfer and output characteristics of several devices. The performance of the 2D FETs is notably improved when using CNT contacts vs traditional 3D metal contacts and the authors attribute this to the tunability of the CNT work function by the back-gate in the contact region. They derive a longitudinal transmission line model to extract resistivity parameters in their devices and claim record-low values for the contact resistance for this aggressively down-scaled contact region.

This work is quite timely, as interest has grown recently in mixed-dimensional 1D/2D heterostructures. These devices have the potential for offering the ultimate device down-scaling paradigm without the need for epitaxial lattice matching. The electrostatic nature of these interfaces has not been explored in detail and there is a distinct lack of experimental reports on certain standard figures of merit, such as the contact resistance. This manuscript addresses some of these issues and proposes a fabrication procedure for making 1D/2D FETs with consistent electrical performance, which is a big advance for the field. In addition, the CNT contacts are shown to improve electron transport quite strongly in intrinsically p-type WSe₂ while preserving high hole mobilities, showing promise for CMOS-like applicability. The novelty in this report is quite high, but some of the conclusions are not supported by the data. Below is a list of outstanding issues that need to be addressed before this paper can be published in Nature Communications:

Major issues:

1. I have a strong impression that the authors buried the lede of their own paper. The key statement of this work appears at the end of the fabrication section on page 3: "The DOS of the CNT is small and almost constant between the two first van Hove singularities, which efficiently suppresses the metal-induced gap states (MIGS) in MoS₂ and thus eliminates the Fermi level pinning." This is the reason why the choice of CNT as a contact material to the 2D semiconductor is a major conceptual advancement described this work. In the abstract and introduction, the authors seem to suggest that gating the contact area allows for improved current injection into the channel, and this is the only reason for better electrical performance. However, this statement would also be true for any 2D semiconductor contact to a TMD, and in some sense is also true for metal contacts. What is unique here is that the band structure and resulting DOS of the semi-metallic CNT suppresses the MIGS-like effect in TMDs. It's precisely the interaction of the 1D DOS of the electrode with the 2D DOS of the channel that allows for the engineering of this efficient contact interface. It's also the reason why mixed-dimensional heterojunctions should be studied more in the future by other groups. The authors should stress this in the introduction of their paper, as right now there seems to be nothing special about the choice of CNT vs other materials—such as graphene or Bi.

2. One claim that seems to crop up throughout the manuscript is that the van der Waals gap between a CNT and a TMD is "obviously" shorter than that between two arbitrary 2D materials. The authors claim that this also enables a higher current flow between the two materials, and affects the tunneling efficiency of electrons. There are several issues with the topic of the vdW gap in this paper that the authors need to address:

a. It is not obvious at all why the vdW gap between a CNT and MoS₂ is smaller than that between graphene and MoS₂. The illustrations shown in Fig. S4 show what looks like VESTA images, with the CNT and graphene structures at some distance away from the top of the MoS₂. However, this drawing can be made arbitrarily in the software to have such a separation and it is unclear what is different physically about these heterostructures. Is the claim here that the carbon atoms in the CNT structure somehow have a different vdW interaction with MoS₂ than the exact same carbon atoms in the graphene lattice? Is this something to do with the hybridization of the carbon orbitals? The authors

need to discuss this in detail before making such a claim. In addition, at the end of the Methods section, the authors state that: "The vacuum distance of at least 12 Å in the z direction was imposed to eliminate the interactions between the periodic images." Again, it is unclear here what this means and how this affects the vdW gap for the calculations.

b. If this difference in the vdW gap is in fact real, can the claim really be made that a separation difference to MoS₂ of 0.36 Å between a CNT and graphene seriously affects charge transport across the interface? Previous works have shown that vdW gaps as large as 1.5 Å can be fully neglected when modelling tunneling transport—see, in particular, the calculation in the Methods section of [1].

c. The authors' own data, in fact, do not support this argument. The vdW gap is clearly visible in the FIB cross-sections in Figs. 1b and S2 in-between the MoS₂ layers, but is hard to discern between the CNT and the MoS₂ in those same images and in the smeared EELS map. I understand it is difficult to obtain clean images of this interface, but right now the TEM images are unconvincing in supporting any arguments on the nature of the vdW gap.

3. It is possible that the SBH extracted for the CNT/MoS₂ interface may be convolved with the SBH of the CNT/Ti interface. Both the CNT/MoS₂ and CNT/Ti junctions in this series resistor are gate-tunable and will show thermionic emission with a flat-band condition for electrons at some positive back-gate voltage (as the CNT is ambipolar and the MoS₂ is n-type). The authors extract and compare the SBH of the CNT and Ti contacts on MoS₂, but they do not provide a reference sample of SBH extraction on a CNT FET contacted with Ti only. This is quite necessary for extracting an absolute SBH value for the mixed-dimensional interface and proving that it is (or isn't) the current-limiting interface in this system. The Ti/CNT resistance is taken into account in the model derivation in later sections—as the R_{nc}^{CNT} and R_Q^{CNT} terms—and somehow plotted out in the inset of Fig. 3c, but there are no direct temperature-dependent flat-band experimental data confirming the SBH extraction for CNT/Ti in the same way as for the others.

4. The final outcome of the derived model is that the 1D/2D contact resistance achieved here is a function of the tunneling resistance (which is a non-trivial function of the separation) and the diameter of the CNT. But the CNT is a cylinder whose contact area with the MoS₂ surface is likely much smaller than the diameter of the whole nanotube. Have the authors measured different diameters of CNTs as contacts to verify this model and to show that the contact resistance does in fact scale as $1/DCNT$? It may be that the results are independent of diameter as only the tip of the CNT touches the MoS₂ surface for all cases.

5. Details of the annealing procedure for removing the S particles from the CNT post-transfer need to be provided. Heat treatment in a sulfur environment may heal S vacancies in the MoS₂ lattice and artificially improve electrical characteristics such as the mobility and transconductance. [2–3] The conditions of temperature, pressure, etc., need to be provided, and a comparison of the MoS₂ electrical characteristics with and without annealing of the CNT contacts post-transfer should be shown to prove that it is not simply the annealing in close proximity to sulfur that improves the electrical performance of the MoS₂ channel.

Minor issues:

6. Several works have been published in the last few years dealing specifically with TMD/CNT heterostructures. The introduction to this paper currently lacks any reference to works such as [4–7] to contextualize recent developments in 1D/2D mixed-dimensional devices and how this work improves on/is different from those previous publications.

7. On page 1: "In the case of traditional 3D metal, the average grain size is approximately equal to the smaller of the width and thickness". It's unclear what the authors are trying to say here. Perhaps a reference could be inserted here to help the reader get a better understanding of the effects of metal grain size on electrical contacts to 2D materials?

8. The authors often refer to their CNTs as being “ultralong”, yet at no point in the paper or SI is it stated what the length of these CNTs actually is, and how this affects the transfer procedure with the W tips.

9. The green I–V curves in Fig. 4e are not described in the figure caption nor discussed in the main text at all.

10. Caption of Fig. 3e says “drops sharply” when it actually increases sharply.

11. Why does the MoS₂ device in Fig. 4b have a negative threshold voltage when all previous MoS₂ devices shown in the paper had a positive threshold voltage?

12. What is the reason for why the output characteristics don’t saturate in Figs. 4c–e? In the case of low-SBH Bi contacts, TMD FETs can enter velocity saturation at much lower V_{ds} values [8]. This might suggest the device behavior is still not channel-dominated here but rather contact-dominated by the CNT/Ti interface, as discussed above.

13. The variables DP, X and β in Fig. 3b are not discussed at all in the text or captions.

14. No scale bars in Fig. S1.

15. The variables described in the figure caption of Fig. S3b are not actually marked on the plot. Caption of subplot d has the wrong formatting.

16. The Ti/MoS₂ interface trend is not plotted at all in Fig. S7f, even though it is mentioned in the caption.

17. In the LTM derivation in the SI, μm is sometimes written as um .

18. Some words in the boldface figure titles are often unnecessarily capitalized.

19. Some language mistakes crop up throughout, e.g., “no experimental exists”, “electrodes that in contact with MoS₂”, “Fermi level of the intrinsic CNT located at its Dirac point”, “and so as the SBH”, etc.

References:

- [1] Liu, Y., Guo, J., Zhu, E., Liao, L., Lee, S.J., Ding, M., Shakir, I., Gambin, V., Huang, Y. and Duan, X., 2018. Approaching the Schottky–Mott limit in van der Waals metal–semiconductor junctions. *Nature*, 557(7707), pp.696–700.
- [2] Yanase, T., Uehara, F., Naito, I., Nagahama, T. and Shimada, T., 2020. Healing sulfur vacancies in monolayer MoS₂ by high-pressure sulfur and selenium annealing: implication for high-performance transistors. *ACS Applied Nano Materials*, 3(10), pp.10462–10469.
- [3] Pak, S., Jang, S., Kim, T., Lim, J., Hwang, J.S., Cho, Y., Chang, H., Jang, A.R., Park, K.H., Hong, J. and Cha, S., 2021. Electrode-Induced Self-Healed Monolayer MoS₂ for High Performance Transistors and Phototransistors. *Advanced Materials*, 33(41), p.2102091.
- [4] Kamaei, S., Saeidi, A., Jazaeri, F., Rassekh, A., Oliva, N., Cavalieri, M., Lambert, B. and Ionescu, A.M., 2020. An Experimental Study on Mixed-Dimensional 1D-2D van der Waals Single-Walled Carbon Nanotube-WSe₂ Hetero-Junction. *IEEE Electron Device Letters*, 41(4), pp.645–648.
- [5] Zhang, J., Wei, Y., Yao, F., Li, D., Ma, H., Lei, P., Fang, H., Xiao, X., Lu, Z., Yang, J. and Li, J., 2017. SWCNT–MoS₂–SWCNT Vertical Point Heterostructures. *Advanced Materials*, 29(7), p.1604469.
- [6] Zhang, J., Cong, L., Zhang, K., Jin, X., Li, X., Wei, Y., Li, Q., Jiang, K., Luo, Y. and Fan, S., 2020. Mixed-dimensional vertical point p–n junctions. *ACS Nano*, 14(3), pp.3181–3189.
- [7] Jadwiszczak, J., Sherman, J., Lynall, D., Liu, Y., Penkov, B., Young, E., Keneipp, R., Drndić, M.,

Hone, J.C. and Shepard, K.L., 2022. Mixed-Dimensional 1D/2D van der Waals Heterojunction Diodes and Transistors in the Atomic Limit. *ACS Nano*, 16(1), pp.1639-1648.
[8] Shen, P.C., Su, C., Lin, Y., Chou, A.S., Cheng, C.C., Park, J.H., Chiu, M.H., Lu, A.Y., Tang, H.L., Tavakoli, M.M. and Pitner, G., 2021. Ultralow contact resistance between semimetal and monolayer semiconductors. *Nature*, 593(7858), pp.211-217

Reviewer #3:

Remarks to the Author:

In this work, a combinational experimental and theoretical study was performed on the transport properties of the semimetallic CNT- 2D semiconductor interface. The authors reported measured 1D-2D contact resistivity value of 10^{-6} Ohm cm², contact resistance value of 50 kOhm*um, Schottky barrier height of 191 meV. They further performed theoretical analysis and predicted that the contact resistivity and contact resistance can be much lower. The semimetal contact (with Bi and Sb) and CNT bundled contact for 2D semiconductors have been reported previously. Because of the bad band alignment between the CNT and MoS2 in this study, A Schottky barrier still exists even though a semimetal contact is used. This basically limits the contact resistance reduction as well as the device performance. The theoretical values predicted in this study did not take into consideration of the contributions from the Schottky barrier, so the prediction also needs to be adjusted. A somewhat new aspect of this study is that the contact length was reduced to ~ 1 nm, which may be important for a future ultimately scaled transistor technology.

Below are my additional questions/comments.

- (1) The authors used the diameter of CNT as the contact length, but because the cross-sectional of CNT is round shape, the actual contact length might be much smaller than the diameter.
- (2) it is not a fair comparison to use Ti/Au contact as the baseline, because Ti/Au contact is well known to be a very bad contact metal for TMDs. The author may need to consider using better candidates, such as Ni or Bi or Au (without Ti) as the control group.
- (3) There is still a Schottky barrier at the CNT-MoS2 interface according to both the experimental results and the theoretical calculations. This means the selected CNT may not be a good choice for low-resistance contacts for TMDs. The authors may need to consider other types of CNTs or 1D semimetallic materials, which have lower work function to start with.
- (4) For the theoretical prediction about the contact resistivity and contact resistance, the authors missed an important term, which is the resistance contributed from the Schottky barrier height. After including this term, the predicted values may become much larger.
- (5) The y-axis of Supplementary Fig. 10 may be mislabeled. Please double check.

Response to Reviewer 1

General Comments: In the manuscript entitled “One-Dimensional Semimetal Contacts to Two-Dimensional Semiconductor”, Li et al., report a novel junction contact between 1D-SWCNT and 2D-MoS₂. This work presents some interesting and well-analyzed results. However, it seems to have a few issues to be considered before publication.

Response: We thank Reviewer 1 very much for his/her kind and valuable comments on our manuscript. The suggestions are very helpful to improve the quality of our manuscript. We would like to address the concern below point-by-point. Supplementary experiments were also carried out to address the comments.

Comment 1: In Figure 1a, how are the authors sure that the sulfur particles were removed after annealing? Since the deposited sulfur particles can affect the charge carrier concentration and electrical properties of the CNTs, it is necessary to show the evidence of the removal of the sulfur particles. It is easy to distinguish the doped CNTs and undoped CNTs by the Raman spectroscopy [S. Suzuki et. al. Carbon 49, 2264-2272 (2011)] and transfer characteristics [Ref. 41]. I suggest to measure the Raman spectra and transfer curves of the CNTs at each step of fabrication processes (Figure 1 a-ii-v).

Response 1: It is known that the sublimation point and melting point of sulfur are 95°C and 112.8°C respectively. Jian et al. [Nanoscale 8, 13437 (2016)] demonstrated that the sulfur particles attached to CNT can be completely removed by heating treatment at 120 °C for 1 min under ambient conditions and thus do not influence the subsequent applications of CNTs. Here, we performed the annealing process under vacuum at 300 °C for 1h, which can achieve more complete desulfurization and form clean and tight van der Waals contact between CNT and TMD. It is technically challenging to measure the transfer curves of the suspended CNTs in fabrication processes ii and iii shown in Figure 1a. To confirm the desulfurization results experimentally, we fabricated a FET with a fresh semimetal CNT channel [PNAS 116, 6586-6593 (2019)]. Supplementary Fig. 5a shows the optical images of the device before sulfur deposition, after sulfur deposition and after annealing. It can be seen that there are no sulfur particles around the device after annealing treatment. Supplementary Fig. 5b presents the corresponding transfer curves, in which the transfer curves of the CNT device before sulfur deposition and after annealing treatment almost coincide. But after sulfur deposition, the transfer characteristics of the CNT became gate-independent due to the doping of sulfur particles. These results indicate that the sulfur particles can be efficiently removed by the annealing treatment. The Raman spectra of CNT in the three states (Supplementary Figure 5c) did not show significant differences, which is consistent with the conclusion in the previous work [Nanoscale 8, 13437 (2016)]. We think that this is because the doping caused by sulfur adhesion on the surface of CNT does not significantly affect the vibration mode of C-C bond. While the doping case discussed by Suzuki et al. [Carbon 49, 2264-2272 (2011)] involves atomic substitution of C with B and N, thus producing a much more significant Raman peak shift. These discussions have been added to the Supplementary Note 1.

Revisions in the manuscript:

1. We have added the details of desulfurization process in the **Fabrication, characterization and DFT calculation of 1D semimetal contact** section in the manuscript.

“The specimen was annealed at 300 °C under 400 mTorr Ar atmosphere for 1 h to remove the sulfur particles while achieving clean and tight mixed-dimensional vdW interface. More comparative experiments have further verified that the sulfur-assisted transfer technique does not affect the transport properties of both SWCNT and MoS₂, which can be found in Supplementary Note 1 and Supplementary Figure 5, 6.”

2. The revised Supplementary Note 1 and Supplementary Figure 5 have been added.

Supplementary Fig. 5 Characterization of the effect of the sulfur deposition and annealing process on CNT. **a**, Optical images of a Ti-contacted CNT FET with semimetal CNT channel before and after sulfur treatment. Scale bar: 4 μ m. **b**, Transfer characteristics of the CNT FET before and after sulfur treatment. **c**, Raman spectra of the CNT before and after sulfur treatment.

Comment 2: As MoS₂ is very vulnerable to the temperature-dependent sulfur vacancy, the authors need to describe the desulfurization process of CNT in more details (including the environments like temperature, atmosphere, and time.). Also, to shoot out the defect issue of the MoS₂, the authors need to provide clear evidences about the state of MoS₂ via a comparative assessment between pre/post-treated MoS₂ samples.

Response 2: We prepared a Ti/Au-contacted MoS₂ FET to check whether the sulfur particles could contribute to the healing of vacancy in MoS₂ by the annealing process. Supplementary Fig. 6a shows the optical images of the MoS₂ FET before sulfur deposition, after sulfur deposition and after annealing. There are no sulfur particles around the device after annealing treatment. The Raman spectra of MoS₂ in the three states (Supplementary Fig. 6c) also show no significant differences. Supplementary Fig. 6b presents the transfer curves of the MoS₂ FET corresponding to the three

states. After sulfur deposition, the threshold voltage shifts positively and the on current of the device decreases significantly, which is induced by the adsorption of sulfur particles. After annealing, the transfer curve of the device almost returns to the initial state. This is because the annealing procedure is carried out under vacuum (400 mTorr, Ar atmosphere), and it does not lead to a significant healing of sulfur vacancy compared with the high-pressure annealing process [3800 Torr, ACS Appl. Nano Mater. 3, 10462–10469 (2020)]. The actual amount of sulfur is decided by the sulfur particles attached on the transferred CNT, which is much less than that in this comparative experiment. Therefore, we can confirm that the sulfur-assisted CNT transfer technique does not affect the electrical properties of MoS₂. These discussions have been added to the Supplementary Note 1.

Revisions in the manuscript:

1. We have added the details of desulfurization process in the **Fabrication, characterization and DFT calculation of 1D semimetal contact** section in the manuscript.

“The specimen was annealed at 300 °C under 400 mTorr Ar atmosphere for 1 h to remove the sulfur particles while achieving clean and tight mixed-dimensional vdW interface. More comparative experiments have further verified that the sulfur-assisted transfer technique does not affect the transport properties of both SWCNT and MoS₂, which can be found in Supplementary Note 1 and Supplementary Figure 5, 6.”

2. The revised Supplementary Note 1 and Supplementary Figure 6 have been added.

Supplementary Fig. 6 Characterization of the effect of the sulfur deposition and annealing process on MoS₂. **a**, Optical images of a Ti-contacted MoS₂ FET before and after sulfur treatment. Scale bar: 4μm. **b**, Transfer characteristics of the MoS₂ FET before and after sulfur treatment. **c**, Raman spectra of the MoS₂ before and after sulfur treatment.

Comment 3: The RBM mode of CNT can span from ~130 cm⁻¹ to ~300 cm⁻¹, depending on its van Hove electronic transition, size, and chirality, etc. However, it is difficult to evaluate this due to the breaking markers. The authors need to roll back the break regime. Like the author’s statements, if

the vdW gap between SWCNT and 2D MoS₂ is so small, it is not suitable to monitor the post-transferred CNT solely. The authors need to proceed the comparative Raman study about both the pre-transfer suspended CNT and the post-transfer CNT in order to evaluate how RBM mode is influenced by the vdW interaction between them. If there occurred some shift due to the interaction, the current radius result on the transferred state can be modified somewhat.

Response 3: We apologize for the incomplete wave number range shown in Figure 1d. The following Figure shows the raw Raman spectra of the CNTs on Si/SiO₂ substrate in range of 100 cm⁻¹ to 400 cm⁻¹ to confirm the RBM mode, which can also be found as Supplementary Fig. 4 in the revised supplementary materials. The RBM peaks around 146 cm⁻¹ verify their single-walled structure and the same chirality of the two SWCNT electrodes. The prominent 303 cm⁻¹ peak comes from the Si/SiO₂ substrate, which is consistent with previous work [Phys. Rev. Lett. 86, 1118 (2001)]. As the signal from the substrate is much prominent than the signal from the SWCNTs, we set the breaking marker at 200 cm⁻¹, which keeps and highlights all the critical information from the CNT. As suggested by Reviewer 1, the RBM mode of SWCNT is affected by many factors, including the type of substrate, chirality of SWCNTs and et al. The structural determination of isolated SWCNT on Si/SiO₂ substrate by resonant Raman scattering has been studied in Phys. Rev. Lett. 86, 1118 (2001). On Si/SiO₂ substrate, the quantitative relationship between the RBM peak and SWCNT diameter can be expressed as $\omega_{RBM} = 248/d_{CNT}$. Therefore, we measured the Raman spectra of CNTs on the Si/SiO₂ substrate in our experiments (the red point in the following Figure b represent the measured CNT positions), from which we can calculate the diameter of CNTs to be 1.7 nm.

Revisions in the manuscript:

1. We have added the location information of the Raman measurement of CNT in the 2nd paragraph of the **Fabrication, characterization and DFT calculation of 1D semimetal contact** section. “Figure 1d displays the Raman spectra of MoS₂ and the two SWCNTs on Si/SiO₂ substrate. More information about the Raman experiments can be found in Supplementary Figure 4.”
2. The diameter calculation of SWCNT is also added in this paragraph. “On Si/SiO₂ substrate, the quantitative relationship between the RBM peak and SWCNT diameter can be expressed as $\omega_{RBM} = 248/d_{CNT}$, and the diameter of CNT can be calculated to be 1.7 nm.”
3. The following figure is included in the revised supplementary information as Supplementary Figure 4.

Supplementary Fig. 4 Raman spectra of the SWCNTs. **a**, The raw Raman spectra of the CNTs on Si/SiO₂ substrate in range of 100 cm⁻¹ to 400 cm⁻¹ to confirm the RBM mode. The 303 cm⁻¹ peaks come from the Si/SiO₂ substrate. **b**, The SEM image of the CNT-contacted MoS₂ FET. The Red point indicates the position of the focused Raman laser beam.

Comment 4: There is no comment on the linear line of low gate bias regime in Figure 2f and Supplementary Figure 7f. Also, Figure S7(f) is not matched with the corresponding caption. Where is the green plot?

Response 4: We are sorry that the linear line of low gate bias regime in Figure 2f may confuse the switching mechanism of the Sm-S junctions and M-S junctions, and it has been deleted. For M-S junctions, it is believed that the critical point where $\Phi_B(V_g)$ deviates from linearity is employed to define the Schottky barrier height (SBH). However, this is not applicable to the Sm-S (CNT-MoS₂) junctions, where the interface SBH is gate-tunable for the tunable work function of semimetal. We have added a detailed discussion on the dependence of the Sm-S interface barrier Φ_B on V_g in the revised Supplementary Note 5. Briefly, the SBH at the Sm-S interface can be efficiently modulated because of the absence of Fermi level pinning and the gate-tunable work function of CNT. The SBH can be reduced to zero and Ohmic contact can thus be achieved. In addition to SBH, the Φ_B modulation is also assisted by the tuning efficiency of MoS₂'s Fermi level, which is more efficient at lower V_g than at higher V_g . The band alignment of the CNT-MoS₂ junction under different gate voltages (Figure 2g) can be confirmed by the CIFS analysis shown in Supplementary Fig. 15e [PNAS 119, e2119016119 (2022)]. We apologize for the absence of the gate-dependent barrier height of the Ti/MoS₂ interface in Supplementary Fig. 7f, and the corresponding data has been added into the Supplementary Fig. 13f in the revised manuscript.

Revisions in the manuscript:

1. The linear lines of low gate bias regime in Figure 2f and Supplementary Figure 7f have been deleted.
2. Supplementary Note 5 has been added to provide a detailed discussion on the dependence of the Sm-S interface barrier Φ_B on V_g .
3. The Supplementary Figure 14 and Supplementary Figure 15e have been added.
4. The data corresponding to the gate-dependent barrier height of the Ti/MoS₂ interface has been added into the revised Supplementary Fig. 13f.

Comment 5: As stated in Ref. 10, the barrier height (Φ_B) can be generally elucidated by the linear combination of work function of CNT, the work function, and affinity of n-type MoS₂ (that is, $\Phi_B = W_{\text{CNT}} - W_{\text{MoS}_2} + \chi_{\text{MoS}_2}$). Assuming that W_{CNT} , W_{MoS_2} , χ_{MoS_2} in the ambient condition have the typical values of $\sim 4.5\text{eV}$, $\sim 4.0\text{eV}$, $\sim 5.1\text{eV}$, respectively, the barrier height at $V_g = 0\text{V}$ is predicted to be roughly as large as $\sim 3.4\text{eV}$. However, Figure 2f shows that Φ_B is at most $\sim 0.6\text{eV}$ at 0V . What is the reason to make such a big difference?

Response 5: We have added the band diagrams of semimetal-semiconductor junctions in Supplementary Fig. 14. As stated by Sze [*Physics of Semiconductor Devices* (Wiley, Hoboken, 2006)] and Yu et al. [Nano Lett. 14, 3055–3063 (2014)], the barrier height (Φ_B) can be expressed as $\Phi_B = W_{\text{Sm}} - \chi_{\text{SC}} + \phi_s$, where W_{Sm} is the work function of (semi)metal, χ_{SC} is the affinity of semiconductor and ϕ_s is the surface potential of semiconductor. For $\phi_s > 0$ (Supplementary Fig. 14a), the expression of Φ_B can be simplified to $\Phi_B = W_{\text{SC}} - \chi_{\text{SC}}$; while for $\phi_s < 0$ (Supplementary

Fig. 14b), the Φ_B is equal to the Schottky barrier height, $\Phi_B = W_{Sm} - \chi_{SC}$. In our experiment, $\phi_{s,MoS_2} > 0$ at $V_g = 0V$ (Supplementary Fig. 15e). Therefore, $\Phi_B = W_{MoS_2} - \chi_{MoS_2} = 5.07 - 4.27 eV = 0.8 eV$, [Appl. Phys. Lett. 103, 053513 (2013); 2D Mater. 4, 015026 (2017)] this value is consistent with our experiment result.

Comment 6: In Figure 2f, the decay rate difference of Φ_B between the two curves seems to be very similar in the high bias regime ($V_g > 15 V$). On the other hand, its bias-induced reduction is pretty remarkable in the lower bias regime ($V_g < 10 V$). Thus, CNT-effect seems to play a more dominant role in lowering the barrier in the low bias regime than in the higher bias regime. It would be better to check if the CNT/MoS₂ sample follows the Square-root-type behavior of SWCNT work function as reported by Yu et al. (Nano Letters 9, 3430 (2009)).

Response 6: We have added a detailed discussion on the dependence of the Sm-S interface barrier Φ_B on V_g in the Supplementary Note 5. In summary, the SBH at the CNT/MoS₂ interface can be efficiently modulated because of the absence of Fermi level pinning and the gate-tunable work function of CNT. The SBH can be reduced to zero and Ohmic contact can thus be achieved. The potential barrier Φ_B at CNT-MoS₂ interface is determined by the energy difference between the work function and affinity of MoS₂ (Supplementary Figure 14a and Figure 15e). That is, the Φ_B modulation is assisted by the tuning efficiency of MoS₂'s Fermi level, which is more efficient at lower V_g than at higher V_g . When the device is in the off state or subthreshold region, the potential barrier Φ_B changes linearly with V_g , as the Fermi level of MoS₂ is located in the band gap and the quantum capacitance of MoS₂ equals 0; when the device is in the on state, the tuning efficiency of the barrier decreases significantly due to the increased quantum capacitance of MoS₂ near the band edge. The Ti contact prepared by electron beam evaporation in our comparative experiments will damage the MoS₂ in the contact area, resulting in additional surface states and may reduce the tuning efficiency of the barrier in the low bias regime.

Since the semimetal SWCNTs have approximately constant DOS between the two first von Hove singularities [Appl. Phys. Lett. 60, 2204 (1992); Rev. Mod. Phys. 79, 677 (2007)], the $\frac{dW_{CNT}}{dV_g}$ should be a constant value. In contrast, graphene has a linear relationship between DOS and energy around the Dirac point, which induces the square-root-type behavior of graphene work function [Yu et al., Nano Letters 9, 3430 (2009)]. Therefore, the V_g -dependent work function of SWCNT and graphene is different.

Revisions in the manuscript:

1. We have made revisions in the **Electrical transport measurements** section.
 - “It can be seen that the Φ_B at CNT-MoS₂ interface changes significantly in the lower V_g regime and the tuning efficiency of the barrier decreases at higher V_g .”
 - “In addition to SBH, the Φ_B modulation is also assisted by the tuning efficiency of MoS₂'s Fermi level, which is more efficient at lower V_g than at higher V_g (See details in Supplementary Note 5).”
2. The added **Supplementary Note 5** is:

Note 5. Gate modulation of the potential barrier at the semimetal-semiconductor (Sm-S) junctions

The band diagrams of Sm-S (n-type) junctions have been sketched in Supplementary Figure 14. The potential barrier at the interface Φ_B can be expressed as:

$$\Phi_B = \begin{cases} SBH, & \phi_s < 0 \\ \phi_n, & \phi_s > 0 \end{cases}$$

where SBH , ϕ_n , $\phi_s = W_{SC} - W_{Sm}$ are Schottky barrier height, the energy difference between the work function and affinity of the semiconductor, the surface potential of the semiconductor, respectively. For $\phi_s < 0$, $\Phi_B = SBH = W_{Sm} - \chi_{SC}$, where W_{Sm} and χ_{SC} are the work function of semimetal and the affinity of semiconductor respectively. For $\phi_s > 0$, $\Phi_B = \phi_n = SBH + \phi_s = W_{Sm} - \chi_{SC} + \phi_s = W_{SC} - \chi_{SC}$, where W_{SC} is the work function of semiconductor.

When an external electrical field is applied, the work functions of both semimetal and semiconductor are tuned following $\frac{1}{e} \frac{dW}{dV_g} = -\frac{1}{e} \frac{d\mu}{dV_g} = -\frac{C_g}{C_Q + C_{it} + C_g}$, where the quantum capacitance C_Q is proportional to the DOS, C_{it} is the interface trap capacitance caused by the surface states, C_g is the gate capacitance of SiO₂ dielectric. At $V_g = -13V \sim 50V$, the work function of CNT is smaller than that of MoS₂ as shown in Supplementary Figure 15e, which means $\phi_s > 0$ and the potential barrier $\Phi_B = W_{MoS_2} - \chi_{MoS_2}$. In this way, the dependence of the barrier on V_g can be expressed as

$$\frac{1}{e} \frac{d\Phi_B}{dV_g} = \frac{1}{e} \frac{dW_{MoS_2}}{dV_g} = -\frac{C_g}{C_{Q,MoS_2} + C_{it} + C_g}.$$

When the Fermi level of MoS₂ is located in the band gap, the $C_{Q,MoS_2} = 0$ and $\frac{1}{e} \frac{d\Phi_B}{dV_g} = -\frac{C_g}{C_{it} + C_g}$.

Therefore, the Φ_B is linearly correlated with V_g and the conduction current in the device increases exponentially (corresponding to the subthreshold region), which is consistent with the low V_g regime ($V_g < 10V$) in Figure 2a and 2f. When the Fermi level of MoS₂ approaches its conduction band edge or the donor level, the C_{Q,MoS_2} increases and the $\frac{d\Phi_B}{dV_g}$ is not a constant anymore and decreases, as shown in the high V_g regime ($V_g > 15V$) in Figure 2f. The device switches to the on state.

3. The Supplementary Figure 14 and Supplementary Figure 15e have been added.

Supplementary Fig. 14 Band diagrams of the semimetal-semiconductor junctions for the surface potential of semiconductor $\phi_s > 0$ (a) and $\phi_s < 0$ (b).

Supplementary Fig. 15e Band diagrams of the CNT/MoS₂ heterostructure under different V_g after junction formation.

Comment 7: As shown in the CNT-contacted sample of Figure 2d, the current in the V_g -region lower than 35 V decreases with decreasing temperature ($V_g < 35$ V insulating), while it increases with temperature for the higher bias region ($V_g > 35$ V metallic). The insulating and metallic properties are related to the barrier height in the corresponding voltage regimes of Figure 2f. That is, the insulating property lasted until the barrier height reaches zero. So does Ti-contacted case in Figure 2d. However, in Figure S7, although the barrier height is nearly 0 eV above 40 V, the current increases with temperature in the overall voltage regime.

Response 7: Many thanks to Reviewer 1 for carefully reading the manuscript and pointing out this contradiction. After carefully checking the experimental records and the previous versions of the manuscript, we find that the data in Supplementary Figure 7d is from a device with a 12-nm-thick MoS₂ channel not a monolayer one. It is the smaller band gap of the thicker MoS₂ that enables the insulator-metal transition at lower V_g . We apologize for mixing up the data from different samples

and the data in the revised Supplementary Figure 13d has been corrected.

Comment 8: In Figure S4, the authors need to explain why the vdW gap of 3.08 Å comes out. Is the vdW gap value independent to the chirality of SWCNT? Can this small gap value be obtained only in the CNT with (5,5)?

Response 8: The heterostructure shown in the Supplementary Figure 7a in the revised manuscript is the most stable structure with the lowest relative energy. The relative energy as a function of vdW gap is plotted in Supplementary Figure 7b. Moreover, the (5,5), (9,0) and (4,6) CNTs with the similar diameter were considered to examine the change of vdW gap. As shown in Supplementary Fig. 8, the calculated vdW gaps of (5,5) CNT/MoS₂, (9,0) CNT/MoS₂ and (4,6) CNT/MoS₂ heterostructures are 3.06, 3.13 and 3.09 Å, respectively, indicating that the smaller vdW gap between SWCNT and MoS₂ can be obtained with different chirality. The smaller vdW gap should come from the 1D geometry structure of CNT, which induces a reduced equilibrium distance between the 1D CNT and 2D MoS₂. This conclusion is in good agreement with previous studies on CNT-MoS₂ [J. Phys. Chem. C 121, 21921–21929 (2017)] and CNT-graphene [New J. Phys. 16 013019 (2014)] heterostructures.

Revisions in the manuscript:

1. We have made revisions in the 3rd paragraph of the **Fabrication, characterization and DFT calculation of 1D semimetal contact** section.
“To present the geometry and electronic structures of the mixed-dimensional heterojunction, density functional theory (DFT) calculations were performed on a (5,5) CNT/MoS₂ heterostructure. The junction atomic structure, differential charge density and electrostatic potential distribution are presented in Figure 1e. Comparative calculations were also conducted on a graphene/MoS₂ heterojunction (Supplementary Figure 7). As shown by the variation of the relative energy as a function of interlayer distance, the most stable structure shows a vdW gap of 3.06 Å for CNT/MoS₂ and 3.42 Å for graphene/MoS₂, respectively. The smaller vdW gap of such 1D-2D junction can be attributed to CNT’s tubular structure.”
2. The relative energy as a function of vdW gap has been added as Supplementary Figure 7b in the revised manuscript.
3. Comparative DFT calculations were performed on the (9,0) CNT/MoS₂ and (4,6) CNT/MoS₂ heterostructures. Supplementary Figure 8 has been added to show the results.

Supplementary Fig. 7 a, Top and side views of the CNT/MoS₂ and graphene/MoS₂ heterostructures. **b**, Relative energy of CNT/MoS₂ and graphene/MoS₂ heterostructures as a function of vdW gap. The CNT/MoS₂ presents obviously smaller vdW gap due to the tubular structure of CNT.

Supplementary Fig. 8 Side views of the as-calculated (5,5) CNT/MoS₂, (9,0) CNT/MoS₂ and (4,6) CNT/MoS₂ heterostructures.

Comment 9: It is required to compare the CNT contacts with graphene contacts. Although the interlayer distances between CNT/MoS₂ and graphene/MoS₂ are slightly different ($\sim 0.36 \text{ \AA}$), Is there a significant difference in the performance of devices with CNT and graphene contacts?

Response 9: We have evaluated the tunneling resistivity at the Ohmic CNT/MoS₂ and Graphene/MoS₂ interfaces. According to the updated DFT calculation results, the CNT/MoS₂ heterojunction achieves Ohmic contact under the positive electric field of 0.2 V/\AA . The

corresponding barrier width w_t and barrier height Φ_t are 1.87 Å and 4.03 eV, respectively. And the calculated tunneling resistivity of electrons is $r_t = 2.882 \times 10^{-9} \Omega \cdot \text{cm}^2$ (Supplementary Note 7). Assume that the difference between the CNT contact and Graphene contact only lies in the barrier width, that is, the barrier width $w_{t,Gr}$ and barrier height $\Phi_{t,Gr}$ are 2.23 Å and 4.03 eV, respectively. The calculated tunneling resistivity of electrons is $r_{t,Gr} = 6.128 \times 10^{-9} \Omega \cdot \text{cm}^2$. Since the tunneling probability of electrons increases exponentially with the barrier width, the separation difference to MoS2 of 0.36 Å makes the tunneling resistivity of the Graphene contact more than double that of CNT contact. In general, the interface contact resistivity as low as $10^{-9} \Omega \cdot \text{cm}^2$ makes CNT and Graphene both good contact materials to 2D semiconductors (in Ohmic mode). In contrast, the CNTs with quasi-1D single crystal structure are the best-performance nano-scale quantum wires, while the graphene owns better current carrying capacity.

Revisions in the manuscript:

1. We have made revisions in the 3rd paragraph of the **Fabrication, characterization and DFT calculation of 1D semimetal contact** section. “Since the tunneling probability of electrons increases exponentially with the barrier width, the smaller vdW gap indicates a smaller tunneling resistance.”
2. We have made revisions in the last paragraph of the **LTLM and extraction of contact resistance** section. “Overall, in short contact limit, the 1D semimetal contact using individual SWCNT is prominent over most of the conventional metal contacts as well as the family of semimetal carbon nanomaterials including graphene and CNT bundles. Such performance can be attributed to the Ohmic contact achieved by gate modulation, the smaller vdW gap between 1D and 2D materials, and the perfect quasi-1D single crystal structure of CNT electrodes.”

Comment 10: The author described the longitudinal transfer length method (LTLM) method to estimate the contact resistance of between CNT/MoS2 channel. The equivalent circuit model was well designed, and the most of parameters used in calculation were also well measured, separately. However, I am concerned about the measurement of $\rho_{CNT}^{on, MoS2}$. The authors mentioned that the $\rho_{CNT}^{on, MoS2}$ was measured at the device in the orange box of Figure 1c. Since the CNTs are placed on the MoS2, the measured current can be influenced by the MoS2. How can the authors be sure that the measured current shows only the resistivity of CNTs?

Response 10: The influence from underlying MoS₂ is neglected in the LTLM, which is reasonable and has been successfully applied in the resistivity comparison methodology to measure the resistivity of the semimetals that in contact with semiconductors [PNAS 119, e2119016119 (2022)]. Intuitively, the resistivity of semiconductors is much larger than that of metals. Furthermore, more evaluations were carried out on such systems. The resistance of a semimetal SWCNT, 1 μm in length, is about 4kΩ. The resistance experienced by the current flowing through MoS₂ is over 1MΩ, which is about 3 orders of magnitude larger than the CNT resistance.

Revisions in the manuscript:

We have made revisions in the 3rd paragraph of the **LTLM and extraction of contact resistance** section. “Since the current flowing through MoS₂ is three orders of magnitude smaller than that in CNT, the influence of MoS₂ can be neglected (electrodes A and E).”

Comment 11: The authors mentioned that the Fermi level pinning at the CNT/MoS2 contact is suppressed by small DOS of the CNT. However, since CNTs were transferred onto MoS2, the interface between the two materials form a van der Waals contact. Therefore, the suppression of pinning might be due to the van Waals gap between the MoS2 and CNT (Nature 557, 696–700, (2018)).

Response 11: From the recent important literatures, it can be concluded that the Fermi level pinning can be induced by some factors, including (i) surface states caused by incomplete covalent bonds, surface dangling bonds or surface reconstructions, (ii) Metal-induced gap states (MIGS) in semiconductor caused by the extended wavefunction from metal, and so on. In 2018, Liu et al., reported that the high-quality vdW integration of metals and 2D semiconductors effectively suppresses the Fermi level pinning, making the SBH at the metal-semiconductor interface close to the Schottky-Mott limit [Nature 557, 696-700, (2018)]. In 2021, Shen et al., further proposed a strategy to reduce the Fermi level pinning as well as contact resistance by suppressing MIGS using semimetal contacts with near-zero DOS at the Fermi level [Nature 593, 211-217 (2021)]. In this work, the suppression of the Fermi level pinning at CNT-MoS2 interface benefits from both the vdW gap and the small DOS of the semimetal CNT.

Revisions in the manuscript:

1. We have made revisions in the 2nd paragraph of the **Introduction** section.
“Furthermore, the density of states (DOS) of the 1D semimetal SWCNT is small and almost constant between the two first van Hove singularities. In such semimetal-semiconductor junctions, the clean and intact vdW interface as well as the suppressed metal-induced gap states (MIGS) in semiconductors can effectively eliminate the Fermi level pinning.”
2. We have made revisions in the 1st paragraph of the **General semimetal CNT contact** section.
“Benefiting from the clean and intact vdW interface as well as the small DOS of CNT, the CNT electrodes perform well as ultra-short contacts for 2D MoS₂ FET.”

Comment 12: In case of the 1D semimetal contact, author explained that injection efficiency of electrons from electrode to channel is greatly improved, and the contact resistance at the short contact limit shall be expressed as $R_c = r_c/D_{CNT}$. According to this expression, it is likely that CNT with a diameter larger than 2nm is better for improving the contact properties. Is there any experimental data using CNT with various diameter?

Response 12: As stated by Reviewer 1, the smaller contact resistance R_c should be obtained using larger diameter CNT for the larger contact length. However, the narrower CNT brings the unique ultrashort contact length as well as the Ohmic contact achieved by gate modulation at the same time, which cannot be reached by conventional top-down method. We have shown the experimental data using CNT with two different diameters in the manuscript. The diameter of the CNT used in the 4L-MoS₂ FET was 1.7 nm and the diameter of the CNT used in the 1L-MoS₂ FET was 1.5 nm (as shown in the AFM images in Supplementary Fig. 3). The calculated interfacial contact resistivity r_c of them tends to be consistent in the Ohmic mode. However, it is a challenge to experimentally verify this principle in a wider range of the CNT diameter. First, it is limited by the CNT materials

we have. The typical diameters of the as-grown ultralong single-walled CNTs are mostly in the range of 1 nm to 2 nm. The CNTs with larger diameters are always multi-walled, which complicates the carrier transport in CNTs. Furthermore, the resistance induced by the scattering at the junctions between the CNT wire on SiO₂ and CNT contacts on MoS₂ cannot be neglected anymore due to the nonlinear dispersion relationship of the multi-walled CNTs [Phys. Rev. Lett. 83, 5098-5101 (1999)]. This will prevent us from accurately extracting the interface contact resistance of CNT/MoS₂. Second, the efficient 1D semimetal contact is contributed by both the 1D geometry and its electronic properties. The wider the diameter, the smaller the energy difference between the two first van Hove singularities. For example, the energy difference of (12, 12)-SWCNT (1.65nm) is 1.5 eV, while that of the (36, 36)-SWCNTs (4.95 nm) is only 0.5 eV [http://www.photon.t.u-tokyo.ac.jp/~maruyama/kataura/1D_DOS.html]. This limits the tunable range of the work function of the semimetal CNTs. Moreover, the DOS of MWCNT is the sum of the DOS of each wall, the large semimetal DOS will reduce the tuning efficiency of the CNT work function as well as the contact performance.

We thank Reviewer 1 again for his/her kind and valuable comments, which have improved the quality of our manuscript.

Response to Reviewer 2

General Comments: The paper by X. Li and colleagues reports on the electrical characteristics of 2D semiconductor FETs, e.g., MoS₂, WSe₂, contacted directly with semi-metallic carbon nanotubes as source and drain electrodes. The authors describe an involved fabrication procedure, perform DFT calculations to estimate the electrostatic nature of the CNT/MoS₂ interface, and report the experimental transfer and output characteristics of several devices. The performance of the 2D FETs is notably improved when using CNT contacts vs traditional 3D metal contacts and the authors attribute this to the tunability of the CNT work function by the back-gate in the contact region. They derive a longitudinal transmission line model to extract resistivity parameters in their devices and claim record-low values for the contact resistance for this aggressively down-scaled contact region.

This work is quite timely, as interest has grown recently in mixed-dimensional 1D/2D heterostructures. These devices have the potential for offering the ultimate device down-scaling paradigm without the need for epitaxial lattice matching. The electrostatic nature of these interfaces has not been explored in detail and there is a distinct lack of experimental reports on certain standard figures of merit, such as the contact resistance. This manuscript addresses some of these issues and proposes a fabrication procedure for making 1D/2D FETs with consistent electrical performance, which is a big advance for the field. In addition, the CNT contacts are shown to improve electron transport quite strongly in intrinsically p-type WSe₂ while preserving high hole mobilities, showing promise for CMOS-like applicability. The novelty in this report is quite high, but some of the conclusions are not supported by the data. Below is a list of outstanding issues that need to be addressed before this paper can be published in Nature Communications:

Response: We thank Reviewer 2 very much for his/her kind and valuable comments on our manuscript. We would like to address the concern below point-by-point. Supplementary experiments were also carried out to address the comments.

Comment 1: I have a strong impression that the authors buried the lede of their own paper. The key statement of this work appears at the end of the fabrication section on page 3: “The DOS of the CNT is small and almost constant between the two first van Hove singularities, which efficiently suppresses the metal-induced gap states (MIGS) in MoS₂ and thus eliminates the Fermi level pinning.” This is the reason why the choice of CNT as a contact material to the 2D semiconductor is a major conceptual advancement described this work. In the abstract and introduction, the authors seem to suggest that gating the contact area allows for improved current injection into the channel, and this is the only reason for better electrical performance. However, this statement would also be true for any 2D semiconductor contact to a TMD, and in some sense is also true for metal contacts. What is unique here is that the band structure and resulting DOS of the semi-metallic CNT suppresses the MIGS-like effect in TMDs. It’s precisely the interaction of the 1D DOS of the electrode with the 2D DOS of the channel that allows for the engineering of this efficient contact interface. It’s also the reason why mixed-dimensional heterojunctions should be studied more in the future by other groups. The authors should stress this in the introduction of their paper, as right now there seems to be nothing special about the choice of CNT vs other materials—such as graphene or Bi.

Response 1: Many thanks to Reviewer 2 for this valuable comment. We have revised the introduction in the manuscript and the band diagrams in Figure 2g, Figure 3d and Figure 4a to address this comment.

Revisions in the manuscript:

1. We have added a discussion to emphasize the uniqueness of the 1D semimetal SWCNT in the 2nd paragraph of the **introduction** section. “Recent progress has shown that they can be assembled with 2D materials to form mixed-dimensional vdW heterostructures with multiple functions. Furthermore, the density of states (DOS) of the 1D semimetal SWCNT is small and almost constant between the two first van Hove singularities. In such semimetal-semiconductor junctions, the clean and intact vdW interface as well as the suppressed metal-induced gap states (MIGS) in semiconductors can effectively eliminate the Fermi level pinning. The small DOS could also enable an efficient modulation on work function of CNTs by external electric field. Therefore, the gate-tunable Schottky barrier height (SBH) at such 1D/2D interface can be predicted. All these specific properties indicate that the individual semimetal SWCNT has great potential as an ultimate scaled contact in 2D electronics.”
2. We have replaced the Dirac cones in Figure 2g, Figure 3d and Figure 4a with the DOS diagram to emphasize the 1D electronic structure of the semimetal SWCNT.

Comment 2: One claim that seems to crop up throughout the manuscript is that the van der Waals gap between a CNT and a TMD is “obviously” shorter than that between two arbitrary 2D materials. The authors claim that this also enables a higher current flow between the two materials, and affects the tunneling efficiency of electrons. There are several issues with the topic of the vdW gap in this paper that the authors need to address:

- a. It is not obvious at all why the vdW gap between a CNT and MoS₂ is smaller than that between graphene and MoS₂. The illustrations shown in Fig. S4 show what looks like VESTA images, with the CNT and graphene structures at some distance away from the top of the MoS₂. However, this drawing can be made arbitrarily in the software to have such a separation and it is unclear what is different physically about these heterostructures. Is the claim here that the carbon atoms in the CNT structure somehow have a different vdW interaction with MoS₂ than the exact same carbon atoms in the graphene lattice? Is this something to do with the hybridization of the carbon orbitals? The authors need to discuss this in detail before making such a claim. In addition, at the end of the Methods section, the authors state that: “The vacuum distance of at least 12 Å in the z direction was imposed to eliminate the interactions between the periodic images.” Again, it is unclear here what this means and how this affects the vdW gap for the calculations.
- b. If this difference in the vdW gap is in fact real, can the claim really be made that a separation difference to MoS₂ of 0.36 Å between a CNT and graphene seriously affects charge transport across the interface? Previous works have shown that vdW gaps as large as 1.5 Å can be fully neglected when modelling tunneling transport—see, in particular, the calculation in the Methods section of [1].
- c. The authors’ own data, in fact, do not support this argument. The vdW gap is clearly visible in the FIB cross-sections in Figs. 1b and S2 in-between the MoS₂ layers, but is hard to discern between the CNT and the MoS₂ in those same images and in the smeared EELS map. I understand it is

difficult to obtain clean images of this interface, but right now the TEM images are unconvincing in supporting any arguments on the nature of the vdW gap.

Response 2: a. The heterostructure shown in the Supplementary Figure 7a in the revised manuscript is the most stable structure with the lowest relative energy. The relative energy as a function of vdW gap is plotted in Supplementary Figure 7b. The smaller vdW gap should come from the 1D geometry structure of CNT, which induces a reduced equilibrium distance between the 1D CNT and 2D MoS₂. Similar to the well-known Leonard-Jones potential ($U(r) = A/r^{12} - B/r^6$) [Proc. R. Soc. A 106, 463–477 (1924); J. Chem. Phys. 125, 084510 (2006)], the van der Waals interactions between layered materials involve both attractive and repulsive forces. The attractive forces include dispersion, dipole–dipole and dipole-induced dipole forces, which is indicated by the item of $1/r^6$. The repulsive force originates from the atomic orbital overlap and is represented by the item of $1/r^{12}$ [Stone, A. *The Theory of Intermolecular Forces*, (Oxford Univ. Press, Oxford, 2013); Nature 567, 23–333 (2019)]. The range of the attractive force is much larger than that of the repulsive force. As shown in the figure bellow, each carbon atom in graphene contributes equivalently to both the attractive force and the repulsive force, as the graphene layer is parallel to the MoS₂ layer. While the curvature of CNT makes less carbon atoms contribute to the repulsive force than to the attractive force for the shorter range of repulsive force, which results in the smaller vdW gap between CNT and MoS₂. This conclusion is in good agreement with previous studies on CNT-MoS₂ [J. Phys. Chem. C 121, 21921–21929 (2017)] and CNT-graphene [New J. Phys. 16 013019 (2014)] heterostructures.

In Methods section, the vacuum distance of at least 12 Å in the z direction was imposed to eliminate the interactions between the periodic images. As shown in the Figure bellow, a $1 \times 3 \times 2$ supercell of CNT/MoS₂ heterostructure is presented. It can be seen that the vacuum distances along y and z direction are 10 and 12.5 Å, respectively, which avoid the interaction between CNTs and CNT/MoS₂ heterostructures of adjacent period. For low dimensional (0D, 1D and 2D) system, this vacuum distance is essential in theoretical calculations.

b. We have evaluated the tunneling resistivity at the Ohmic CNT/MoS₂ and Graphene/MoS₂ interfaces. According to the updated DFT calculation results, the CNT/MoS₂ heterojunction achieves Ohmic contact under the positive electric field of 0.2 V/Å. The corresponding barrier width w_t and barrier height Φ_t are 1.87 Å and 4.03 eV, respectively. And the calculated tunneling resistivity of electrons is $r_t = 2.882 \times 10^{-9} \Omega \cdot \text{cm}^2$. Assume that the difference between the CNT contact and Graphene contact only lies in the barrier width, that is, the barrier width $w_{t,Gr}$ and barrier height $\Phi_{t,Gr}$ are 2.23 Å and 4.03 eV, respectively. The calculated tunneling resistivity of electrons is $r_{t,Gr} = 6.128 \times 10^{-9} \Omega \cdot \text{cm}^2$. Since the tunneling probability of electrons decreases exponentially with the barrier width, the separation difference to MoS₂ of 0.36 Å makes the tunneling resistivity of the Graphene contact more than double that of CNT contact. In 2018, Liu et al., reported that the high-quality vdW integration of metals and 2D semiconductors effectively suppresses the Fermi level pinning, making the SBH at the metal-semiconductor interface close to the Schottky-Mott limit [Nature 557, 696-700, (2018)]. The Schottky barrier exist at the metal/semiconductor interface. Compared with the contact resistance caused by the Schottky barrier ($10^{-5} \sim 10^{-3} \Omega \cdot \text{cm}^2$), the electron tunneling resistance induced by the vdW gap is indeed negligible ($10^{-9} \Omega \cdot \text{cm}^2$). This indicates that Ohmic contact and vdW contact are the two essential factors to achieve high performance contact to 2D semiconductors.

c. The STEM and EELS images in Figure 1b and Supplementary Fig. 2 are used to confirm the CNT/MoS₂ heterostructure, which is the best result we have got after many attempts. As stated by Reviewer 2, it is challenging to obtain the atomically sharp STEM image of the CNT-MoS₂ interface experimentally, due to the poor electron beam tolerance and the single layered nanostructure of CNT. Therefore, we further use DFT calculation to effectively analyze the mixed-dimensional heterostructure.

Revisions in the manuscript:

1. We have made revisions in the 3rd paragraph of the **Fabrication, characterization and DFT calculation of 1D semimetal contact** section.

“To present the geometry and electronic structures of the mixed-dimensional heterojunction, density functional theory (DFT) calculations were performed on a (5,5) CNT/MoS₂ heterostructure. The junction atomic structure, differential charge density and electrostatic potential distribution are presented in Figure 1e. Comparative calculations were also conducted

on a graphene/MoS₂ heterojunction (Supplementary Figure 7). As shown by the variation of the relative energy as a function of interlayer distance, the most stable structure shows a vdW gap of 3.06 Å for CNT/MoS₂ and 3.42 Å for graphene/MoS₂, respectively. The obviously small vdW gap of such 1D-2D junction can be attributed to CNT's tubular structure. Since the tunneling probability of electrons increases exponentially with the barrier width, the smaller vdW gap indicates a smaller tunneling resistance."

2. The relative energy as a function of vdW gap has been added as Supplementary Figure 7b in the revised manuscript.

Supplementary Fig. 7 a, Top and side views of the CNT/MoS₂ and graphene/MoS₂ heterostructures. **b**, Relative energy of CNT/MoS₂ and graphene/MoS₂ heterostructures as a function of vdW gap. The CNT/MoS₂ presents obviously smaller vdW gap due to the tubular structure of CNT.

Comment 3: It is possible that the SBH extracted for the CNT/MoS₂ interface may be convolved with the SBH of the CNT/Ti interface. Both the CNT/MoS₂ and CNT/Ti junctions in this series resistor are gate-tunable and will show thermionic emission with a flat-band condition for electrons at some positive back-gate voltage (as the CNT is ambipolar and the MoS₂ is n-type). The authors extract and compare the SBH of the CNT and Ti contacts on MoS₂, but they do not provide a reference sample of SBH extraction on a CNT FET contacted with Ti only. This is quite necessary for extracting an absolute SBH value for the mixed-dimensional interface and proving that it is (or isn't) the current-limiting interface in this system. The Ti/CNT resistance is taken into account in the model derivation in later sections—as the R_{nc}^{CNT} and R_Q^{CNT} terms—and somehow

plotted out in the inset of Fig. 3c, but there are no direct temperature-dependent flat-band experimental data confirming the SBH extraction for CNT/Ti in the same way as for the others.

Response 3: As we know, the CNTs with different helicity can exhibit metallic or semiconducting properties [Appl. Phys. Lett. 60, 2204-2206 (1992); Reviews of Modern Physics 79, 677-732 (2007)], and the CNTs used in our experiment are all the metallic ones. Metallic CNTs are semimetals with no band gap and are always conductive. There is no Schottky barrier at the Ti/CNT interface. The band diagrams of the Ti/CNT junctions before and after contact formation have been added as Supplementary Figure 12c. Experimentally, we have supplemented the temperature-dependent transfer characteristics and output characteristics of the CNT devices in Supplementary Fig. 12a and 12b. It can be seen that as the temperature decreases, the conduction current of the device increases, showing good metallicity and indicating that there is no potential barrier at the Ti/CNT interface. In the CNT-contacted MoS₂ transistors, the contact resistance between Ti and CNT is $R_1 = 2R_{nc}^{CNT} + R_Q^{CNT}$, where $R_{nc}^{CNT} \sim 2k\Omega$ is the interfacial resistance caused by the imperfect contact between Ti and CNT, $R_Q^{CNT} = 6.5 k\Omega$ is the quantum resistance determined by the number of conductive channels of CNT [Nature nanotechnology 5, 858-862 (2010)]. Thus, the $R_1 \sim 10.5k\Omega$. In contrast, the sheet resistivity of the 4L-MoS₂ was measured as $\rho^{2D} = 120 k\Omega \cdot \square^{-1}$ at $V_g = 50V$, and the channel resistance of the device ($L=9.2\mu m$, $W=3\mu m$) equals $368 k\Omega$. Therefore, the contact resistance at the Ti/CNT interface is much smaller and it is not the current-limiting interface in this system.

Revisions in the manuscript:

The Supplementary Figure 12 has been added.

Supplementary Fig. 12 Temperature-dependent electrical transport properties of the semimetal SWCNT. The temperature-dependent transfer curves (a) and output curves (b) of the semimetal SWCNT. Inset is the SEM image of the CNT device. Scale bar: $2\mu m$. The conductivity of semimetal SWCNT increases with the temperature decreases, showing good metallicity and indicating that there is no potential barrier at the metal/CNT interface. c, Band diagrams of the metal/CNT junctions before and after contact formation.

Comment 4: The final outcome of the derived model is that the 1D/2D contact resistance achieved here is a function of the tunneling resistance (which is a non-trivial function of the separation) and the diameter of the CNT. But the CNT is a cylinder whose contact area with the MoS₂ surface is likely much smaller than the diameter of the whole nanotube. Have the authors measured different diameters of CNTs as contacts to verify this model and to show that the contact resistance does in fact scale as $1/DCNT$? It may be that the results are independent of diameter as only the tip of the CNT touches the MoS₂ surface for all cases.

Response 4: Considering the feature size of the contact geometry, we default that the contact length of the 1D semimetal contact is equal to the diameter of CNT. We have defined it in the revised **LTLM and extraction of contact resistance** section. “The contact length l_c is defined as the diameter of CNT (the feature size of contact geometry).” However, the tubular structure of CNT makes things more complicated, as the actual contact length is probably smaller than the CNT diameter. It is an important scientific issue and should continue to be studied in depth. In experiment, we have shown the experimental data using CNT with two different diameters in the manuscript. The diameter of the CNT used in the 4L-MoS₂ FET was 1.7 nm and the diameter of the CNT used in the 1L-MoS₂ FET was 1.5 nm (as shown in the AFM images in Supplementary Fig. 3). The calculated interfacial contact resistivity r_c of them tends to be consistent in the Ohmic mode. However, it is a challenge to experimentally verify this principle in a wider range of the CNT diameter. First, it is limited by the CNT materials we have. The CNTs with larger diameters are always multi-walled, which complicates the carrier transport in CNTs. Furthermore, the resistance induced by the scattering at the junctions between the CNT wire on SiO₂ and CNT contacts on MoS₂ cannot be neglected anymore due to the nonlinear dispersion relationship of the multi-walled CNTs [Phys. Rev. Lett. 83, 5098-5101 (1999)]. This will prevent us from accurately extracting the interface contact resistance of CNT/MoS₂. On the other hand, the efficient 1D semimetal contact is contributed by both the 1D geometry and its electronic properties. The larger the diameter, the smaller the energy difference between the two first van Hove singularities. For example, the energy difference of (12, 12)-SWCNT (1.65nm) is 1.5 eV, while that of the (36, 36)-SWCNTs (4.95 nm) is only 0.5 eV [http://www.photon.t.u-tokyo.ac.jp/~maruyama/kataura/1D_DOS.html]. This limits the tunable range of the work function of the semimetal CNTs. Moreover, the DOS of MWCNT is the sum of the DOS of each wall, the large semimetal DOS will reduce the tuning efficiency of the CNT work function as well as the contact performance.

Comment 5: Details of the annealing procedure for removing the S particles from the CNT post-transfer need to be provided. Heat treatment in a sulfur environment may heal S vacancies in the MoS₂ lattice and artificially improve electrical characteristics such as the mobility and transconductance. [2–3] The conditions of temperature, pressure, etc., need to be provided, and a comparison of the MoS₂ electrical characteristics with and without annealing of the CNT contacts post-transfer should be shown to prove that it is not simply the annealing in close proximity to sulfur that improves the electrical performance of the MoS₂ channel.

Response 5: We prepared a Ti/Au-contacted MoS₂ FET to check whether the sulfur particles could contribute to the healing of vacancy in MoS₂ by the annealing process. Supplementary Fig. 6a shows the optical images of the MoS₂ FET before sulfur deposition, after sulfur deposition and after annealing. There are no sulfur particles around the device after annealing treatment. The Raman spectra of MoS₂ in the three states (Supplementary Fig. 6c) also show no significant differences. Supplementary Fig. 6b presents the transfer curves of the MoS₂ FET corresponding to the three states. After sulfur deposition, the threshold voltage shifts positively and the on current of the device decreases significantly, which is induced by the adsorption of sulfur particles. After annealing, the transfer curve of the device almost returns to the initial state. This is because the annealing procedure

is carried out under vacuum (400 mTorr, Ar atmosphere), and it does not lead to a significant healing of sulfur vacancy compared with the high-pressure annealing process [3800 Torr, ACS Appl. Nano Mater. 3, 10462–10469 (2020)]. The actual amount of sulfur is decided by the sulfur particles attached on the transferred CNT, which is much less than that in this comparative experiment. Therefore, we can confirm that the sulfur-assisted CNT transfer technique does not affect the electrical properties of MoS₂. These discussions have been added to the Supplementary Note 1.

It is difficult to fabricate a CNT-contacted MoS₂ FET without annealing, as the annealing treatment is also necessary to form an intensive binding force between CNT and the substrate. Without annealing, the contact between CNT with sulfur particles and the chip is poor and the CNT will slip away in the spin coating process, resulting in the failure of device fabrication.

Revisions in the manuscript:

1. We have added the details of desulfurization process in the **Fabrication, characterization and DFT calculation of 1D semimetal contact** section in the manuscript.

“The specimen was annealed at 300 °C under 400 mTorr Ar atmosphere for 1 h to remove the sulfur particles while achieving clean and tight mixed-dimensional vdW interface. More comparative experiments have further verified that the sulfur-assisted transfer technique does not affect the transport properties of both SWCNT and MoS₂, which can be found in Supplementary Note 1 and Supplementary Figure 5, 6.”

2. The revised Supplementary Note 1 and Supplementary Figure 6 have been added.

Supplementary Fig. 6 Characterization of the effect of the sulfur deposition and annealing process on MoS₂. **a**, Optical images of a Ti-contacted MoS₂ FET before and after sulfur treatment. Scale bar: 4μm. **b**, Transfer characteristics of the MoS₂ FET before and after sulfur treatment. **c**, Raman spectra of the MoS₂ before and after sulfur treatment.

Comment 6: Several works have been published in the last few years dealing specifically with

TMD/CNT heterostructures. The introduction to this paper currently lacks any reference to works such as [4–7] to contextualize recent developments in 1D/2D mixed-dimensional devices and how this work improves on/is different from those previous publications.

Response 6: Many thanks to Reviewer 2 for this valuable comment. We have revised the 2nd paragraph of the introduction in the manuscript and added the references on 1D/2D mixed-dimensional devices.

“Recent progress has shown that they can be assembled with 2D materials to form mixed-dimensional vdW heterostructures with multiple functions³⁹⁻⁴².”

Comment 7: On page 1: “In the case of traditional 3D metal, the average grain size is approximately equal to the smaller of the width and thickness”. It’s unclear what the authors are trying to say here. Perhaps a reference could be inserted here to help the reader get a better understanding of the effects of metal grain size on electrical contacts to 2D materials?

Response 7: The sentence “In the case of traditional 3D metal, the average grain size is approximately equal to the smaller of the width and thickness” was used to indicate that the average metal grain size is generally determined by the minimum feature size of the metal wire. This meaning of this sentence has been included in the next sentence, so it has been deleted in the revised manuscript. Detailed studies on the resistivity increase induced by metal domainization have been given by the literatures cited as references 24-27.

Comment 8: The authors often refer to their CNTs as being “ultralong”, yet at no point in the paper or SI is it stated what the length of these CNTs actually is, and how this affects the transfer procedure with the W tips.

Response 8: We apologize for the lack of detailed description on the ultralong CNTs. The length the CNTs in this work is about 1 cm, which can reach over 10 cm in literatures [J. Phys. Chem. B 110, 11103–11109 (2006); Nano Lett. 9, 3137–3141 (2009)]. We also added the Supplementary Video to track the same centimeter-long CNT on the SiN_x/Si substrate, where each trench is 200 μm long and every two trenches are 1.6 mm apart. The 200μm-long freestanding section can be picked up and transferred under optical microscope with two W tips. Such ultralong CNT can maintain the same chirality within the centimeter length, which enables us to obtain two same CNT segments at different trenches, thus realizing the symmetrical 1D semimetal contacts to the 2D channels.

Revisions in the manuscript:

1. We have added description on the CNT length and cited the supplementary information in the **Fabrication, characterization and DFT calculation of 1D semimetal contact** section.
2. We have added description in the **Synthesis of ultralong CNT** section in Methods.
“The as-grown centimeter-long CNTs with suspending sections over trenches can be observed in Supplementary Figure 1 and Supplementary Video.”
3. We have also added the Supplementary Video to track the same centimeter-long CNT on the SiN_x/Si substrate.

Comment 9: The green I–V curves in Fig. 4e are not described in the figure caption nor discussed in the main text at all.

Response 9: The green I–V curves in Fig. 4e represent the results at negative V_g . That is, as V_g is swept from -5V to 5V, the CNT/WSe₂ FET exhibit good ambipolar characteristics, and the conduction current decreases first and then increases. To improve the intuitiveness of Figure 4c–e, we have revised the labels and changed the color of arrows.

Revisions in the manuscript:

1. We have added description in the **General semimetal CNT contact** section. “The arrows with gradient color indicate the direction of V_g sweeping.”
2. We have revised the labels and changed the color of arrows. The figure caption has also been revised.

Figure 4 c-e, Output characteristics of MoS₂ (c), WS₂ (d) and WSe₂ (e) FETs, showing V_g -dependent I–V relationship and on-state current of several microampere. The arrows with gradient color indicate the direction of V_g sweeping. The V_g step is 1 V.

Comment 10: Caption of Fig. 3e says “drops sharply” when it actually increases sharply.

Response 10: We apologize for the descriptive error here, which has been corrected.

Comment 11: Why does the MoS₂ device in Fig. 4b have a negative threshold voltage when all previous MoS₂ devices shown in the paper had a positive threshold voltage?

Response 11: The positive or negative value of the threshold voltage depends on the work function difference between the gate material and the semiconductor channel. If the work function of gate material is larger, the threshold voltage should be a negative value. If the work function of semiconductor is larger, the versus applies. The device in Figure 2a and Figure 4b used N-type Si and Au as the gate electrodes, whose work function are 4.2 eV and 5.1 eV, respectively. In comparison, the work function of the N-doped MoS₂ is somewhere in between. Therefore, the device in Fig. 4b has a negative threshold voltage while the one in Fig. 2a has a positive value.

Revisions in the manuscript:

We have added description in the **General semimetal CNT contact** section. “The negative

threshold voltage of this MoS₂ device compared with the one in Fig. 2a should be attributed to their different gate materials.”

Comment 12: What is the reason for why the output characteristics don't saturate in Figs. 4c–e? In the case of low-SBH Bi contacts, TMD FETs can enter velocity saturation at much lower V_{ds} values [8]. This might suggest the device behavior is still not channel-dominated here but rather contact-dominated by the CNT/Ti interface, as discussed above.

Response 12: The unsaturated output characteristics in Figs. 4c–e should be attributed to the μm -long channel of our device rather than the CNT/Ti interface. The transfer operation was carried out under optical microscope. Thus, the CNT-contacted MoS₂ FETs prepared in this work have channel lengths of several microns, which limits the electric field strength in the channel. In addition, as response to the comment 3, there is no Schottky barrier at the Ti/CNT interface. As reported by Shen et al. [Nature 593, 211-217 (2021)], the Bi-contacted MoS₂ FET with 120nm channel length enters velocity saturation at $V_d=1\text{V}$, while the one with $1\mu\text{m}$ channel length does not. Similarly, the In-contacted MoS₂ FET [Nature 568, 70-74 (2019)] is still not saturated at $V_d=3\text{V}$.

Comment 13: The variables DP, X and β in Fig. 3b are not discussed at all in the text or captions.

Response 13: We apologize for the missing description on the feature points appeared in Figure 3b. These feature points were used to analyze the CIFS in semimetal CNT (Supplementary Note 4) to determine the band diagram of the mixed-dimensional CNT/MoS₂ heterojunction (Figure 2g and supplementary Figure 15e).

Revisions in the manuscript:

We revised the figure caption of Figure 3b to include the above information. “The feature points β , X, DP and DP' are used for the CIFS analysis in Supplementary Note 4.”

Comment 14: No scale bars in Fig. S1.

Response 14: We apologize for the absence of scale bars in Supplementary Fig. 1. The scale bars have been added.

Comment 15: The variables described in the figure caption of Fig. S3b are not actually marked on the plot. Caption of subplot d has the wrong formatting.

Response 15: The variables described in the figure caption of Supplementary Fig. 3b have been added to the plot. The “spectrum” in the Caption of Supplementary Fig. 3d has been revised as “spectra”.

Comment 16: The Ti/MoS₂ interface trend is not plotted at all in Fig. S7f, even though it is mentioned in the caption.

Response 16: We apologize for the absence of the gate-dependent barrier height of the Ti/MoS₂ interface, the corresponding data has been added in the revised Supplementary Fig. 13f.

Comment 17: In the LTLM derivation in the SI, μm is sometimes written as um .

Response 17: All the “um” in the LTLM derivation in the Supplementary Information have been revised as “ μm ”.

Comment 18: Some words in the boldface figure titles are often unnecessarily capitalized.

Response 18: The unnecessarily capitalized words in the boldface figure titles have been changed to lowercase.

Comment 19: Some language mistakes crop up throughout, e.g., “no experimental exists”, “electrodes that in contact with MoS₂”, “Fermi level of the intrinsic CNT located at its Dirac point”, “and so as the SBH”, etc.

Response 19: We apologize for the language mistakes, and the mistakes have been revised.

Revisions in the manuscript:

1. “no experimental exists” has been revised as “no experimental result exists”.
2. “electrodes that in contact with MoS₂” has been revised as “electrodes in contact with MoS₂”
3. The sentence “the Fermi level of the intrinsic CNT located at its Dirac point (Figure 2g) while the Fermi level of MoS₂ was modulated into the band gap.” has been deleted from the manuscript.
4. “the work function of CNT decreases and so as the SBH” has been revised as “the work function of CNT as well as the SBH significantly decreases”.

We thank Reviewer 2 again for his/her kind and valuable comments, which have improved the quality of our manuscript.

Response to Reviewer 3

General Comments: In this work, a combinational experimental and theoretical study was performed on the transport properties of the semimetallic CNT- 2D semiconductor interface. The authors reported measured 1D-2D contact resistivity value of 10^{-6} Ohm cm², contact resistance value of 50 kOhm*um, Schottky barrier height of 191 meV. They further performed theoretical analysis and predicted that the contact resistivity and contact resistance can be much lower. The semimetal contact (with Bi and Sb) and CNT bundled contact for 2D semiconductors have been reported previously. Because of the bad band alignment between the CNT and MoS₂ in this study, A Schottky barrier still exists even though a semimetal contact is used. This basically limits the contact resistance reduction as well as the device performance. The theoretical values predicted in this study did not take into consideration of the contributions from the Schottky barrier, so the prediction also needs to be adjusted. A somewhat new aspect of this study is that the contact length was reduced to ~ 1 nm, which may be important for a future ultimately scaled transistor technology.

Response: We thank Reviewer 3 very much for his/her valuable comments on our manuscript. We would like to address the concern below point-by-point. Supplementary experiments were also carried out to address the comments.

We apologize for the unclear statement that the Ohmic contact can be formed between CNT and MoS₂ under positive V_g . The Schottky barrier height at the Sm-S interface can be efficiently modulated because of the absence of Fermi level pinning and the gate-tunable work function of CNT. The SBH can be reduced to zero and Ohmic contact can thus be achieved. There might be two issues in previous manuscript that may cause misunderstanding and we have made relevant revisions.

First, the PDOS plot in Figure 1f shows that there is an n-type Schottky barrier between CNT and MoS₂ without applying an external electric field. We have supplemented more DFT calculations with an applied electric field, and the band structures of the CNT/MoS₂ heterojunction have been shown in Supplementary Fig. 9. It can be seen that, consistent with our experimental results, the CNT/MoS₂ heterojunction achieves Ohmic contact under the positive electric field of 0.2 V/Å.

Second, for M-S junctions, it is believed that the critical point where $\Phi_B(V_g)$ deviates from linearity is employed to define the Schottky barrier height (SBH). However, this is not applicable to the Sm-S (CNT-MoS₂) junctions, where the interface SBH is gate-tunable. We have added a detailed discussion on the dependence of the Sm-S interface barrier Φ_B on V_g in the revised Supplementary Note 5.

Revisions in the manuscript:

1. We have made revisions in the 3rd paragraph of the **Fabrication, characterization and DFT calculation of 1D semimetal contact** section.

“Figure 1f shows projected density of states (PDOS) of the heterostructure. **The small and nearly constant DOS endows the CNT with gate-tunable work function, and efficiently suppresses the MIGS in MoS₂. Therefore, the gate-tunable SBH at the CNT/MoS₂ interface can be predicted. Without external electrical field, the Fermi level of CNT is 0.35 eV lower than the conduction band (CB) edge of MoS₂, indicating an n-type Schottky barrier. While with applied electrical field of 0.2 V/Å, the Ohmic contact can be achieved (Supplementary Figure 9), the**

corresponding width w_t and height Φ_t of the potential barrier are calculated to be 1.87 Å and 4.03 eV, respectively (Supplementary Figure 10). This contributes to a small tunnelling resistivity of $2.882 \times 10^{-9} \Omega \cdot \text{cm}^2$ (see details in Supplementary Note 8). These results present the potential of 1D semimetal contact for 2D FETs.”

2. We have supplemented the DFT calculations on the (5,5) CNT/MoS₂ heterostructure with an applied electric field. The results have been added in the Supplementary Figure 9.

Supplementary Fig. 9 Unfolded band structures of (a) MoS₂ and (b) CNT in the CNT/MoS₂ heterostructure under different external electric fields.

3. We have made revisions to the 2nd paragraph of the **Electrical transport measurements** section. “In addition to SBH, the Φ_B modulation is also assisted by the tuning efficiency of MoS₂’s Fermi level, which is more efficient at lower V_g than at higher V_g (See details in Supplementary Note 5).”
4. We have added the **Supplementary Note 5**:

Note 5. Gate modulation of the potential barrier at the semimetal-semiconductor (Sm-S) junctions

The band diagrams of Sm-S (n-type) junctions have been sketched in Supplementary Figure 14. The potential barrier at the interface Φ_B can be expressed as:

$$\Phi_B = \begin{cases} SBH, & \phi_s < 0 \\ \phi_n, & \phi_s > 0 \end{cases}$$

where SBH , ϕ_n , $\phi_s = W_{SC} - W_{Sm}$ are Schottky barrier height, the energy difference between the work function and affinity of the semiconductor, the surface potential of the semiconductor, respectively. For $\phi_s < 0$, $\Phi_B = SBH = W_{Sm} - \chi_{SC}$, where W_{Sm} and χ_{SC} are the work function of semimetal and the affinity of semiconductor respectively. For $\phi_s > 0$, $\Phi_B = \phi_n = SBH + \phi_s = W_{Sm} - \chi_{SC} + \phi_s = W_{SC} - \chi_{SC}$, where W_{SC} is the work function of semiconductor.

When an external electrical field is applied, the work functions of both semimetal and semiconductor are tuned following $\frac{1}{e} \frac{dW}{dV_g} = -\frac{1}{e} \frac{d\mu}{dV_g} = -\frac{C_g}{C_Q + C_{it} + C_g}$, where the quantum capacitance

C_Q is proportional to the DOS, C_{it} is the interface trap capacitance caused by the surface states, C_g is the gate capacitance of SiO₂ dielectric. At $V_g = -13V \sim 50V$, the work function of CNT is smaller than that of MoS₂ as shown in Supplementary Figure 15e, which means $\phi_s > 0$ and the potential barrier

$\Phi_B = W_{\text{MoS}_2} - \chi_{\text{MoS}_2}$. In this way, the dependence of the barrier on V_g can be expressed as

$$\frac{1}{e} \frac{d\Phi_B}{dV_g} = \frac{1}{e} \frac{dW_{\text{MoS}_2}}{dV_g} = - \frac{C_g}{C_{Q,\text{MoS}_2} + C_{it} + C_g}.$$

When the Fermi level of MoS₂ is located in the band gap, the $C_{Q,\text{MoS}_2} = 0$ and $\frac{1}{e} \frac{d\Phi_B}{dV_g} = - \frac{C_g}{C_{it} + C_g}$.

Therefore, the Φ_B is linearly correlated with V_g and the conduction current in the device increases exponentially (corresponding to the subthreshold region), which is consistent with the low V_g regime ($V_g < 10\text{V}$) in Figure 2a and 2f. When the Fermi level of MoS₂ approaches its conduction band edge or the donor level, the C_{Q,MoS_2} increases and the $\frac{d\Phi_B}{dV_g}$ is not a constant anymore and

decreases, as shown in the high V_g regime ($V_g > 15\text{V}$) in Figure 2f. The device switches to the on state.

5. We have added the Supplementary Figure 14 and Supplementary Figure 15e.

Supplementary Fig. 14 Band diagrams of the semimetal-semiconductor junctions for the surface potential of semiconductor $\phi_s > 0$ (a) and $\phi_s < 0$ (b).

Supplementary Fig. 15e Band diagrams of the CNT/MoS₂ heterostructure under different V_g after junction formation.

Comment 1: The authors used the diameter of CNT as the contact length, but because the cross-

sectional of CNT is round shape, the actual contact length might be much smaller than the diameter.

Response 1: We agree with Reviewer 3. The tubular structure of CNT makes the actual contact length smaller than the CNT diameter. It is an important scientific issue and should continue to be studied in depth. In this work, we use the CNT diameter as the contact length mainly for the following two reasons. First, considering the feature size of the contact geometry, we default that the contact length of the 1D semimetal contact is equal to the diameter of CNT. Second, if we use the CNT diameter as the contact length to calculate the interface contact resistivity by the Equation 2, the as-calculated r_c will be larger than the real case. Therefore, the definition on interface contact resistivity is conservative.

Comment 2: it is not a fair comparison to use Ti/Au contact as the baseline, because Ti/Au contact is well known to be a very bad contact metal for TMDs. The author may need to consider using better candidates, such as Ni or Bi or Au (without Ti) as the control group.

Response 2: Thank you for your suggestion. Our devices are composed of 1D CNT and 2D TMD, and Ti/Au is a good choice for both of them. We have tried to supplemented experiments to compare the SWCNT contact with more other metal contacts. Limited by our device fabrication equipment, we tried Ni/Au and Au, and the results were presented in Supplementary Figure 19. The MoS₂ FETs with CNT contacts (<2nm in contact length) exhibit close on-state current and larger on/off ratio compared to the devices using Ni and Au contacts (3 μm in contact length). In addition, we have compared the contact resistance R_c of CNT with the state-of-the-art contacts in literatures (Bi, Au, In, Ni and graphene) in Figure 3e.

Revisions in the manuscript:

The Supplementary Figure 19 has been added.

Supplementary Fig. 19 Comparison of the 1D semimetal contacts and Ni, Au contacts. Transfer

characteristics (a) and optical image (b) of a MoS₂ FET with SWCNT (red, electrodes 1 and 2) and Ni (blue, electrodes A and B) contacts on P-type Si/300 nm SiO₂ substrate. b, Transfer characteristics (c) and optical image (d) of a MoS₂ FET with SWCNT (red, electrodes 1 and 2) and Au (yellow, electrodes A and B) contacts. Scale bar: 6 μm.

Comment 3: There is still a Schottky barrier at the CNT-MoS₂ interface according to both the experimental results and the theoretical calculations. This means the selected CNT may not be a good choice for low-resistance contacts for TMDs. The authors may need to consider other types of CNTs or 1D semimetallic materials, which have lower work function to start with.

Response 3: As response to the general comment, The Schottky barrier height at the Sm-S interface can be efficiently modulated because of the absence of Fermi level pinning and the gate-tunable work function of CNT. The SBH can be reduced to zero and Ohmic contact can thus be achieved. This conclusion can be further confirmed by the DFT calculation results (Supplementary Figure 9) and the electrical transport measurements in Figure 2 and Supplementary Figure 13.

Revisions in the manuscript:

1. We have supplemented the DFT calculations on the (5,5) CNT/MoS₂ heterostructure with an applied electric field. Descriptions have been added in in the 3rd paragraph of the **Fabrication, characterization and DFT calculation of 1D semimetal contact** section. The calculation results have been added in the Supplementary Figure 9.
2. We have made revisions to the 2nd paragraph of the **Electrical transport measurements** section and added the Supplementary Note 5 to give a detailed discussion on the gate modulation of the potential barrier at the Sm-S junctions. The Supplementary Figure 14 and Supplementary Figure 15e have also been added.

Comment 4: For the theoretical prediction about the contact resistivity and contact resistance, the authors missed an important term, which is the resistance contributed from the Schottky barrier height. After including this term, the predicated values may become much larger.

Response 4: Many thanks to Reviewer 3 for pointing out our negligence in calculating the tunneling resistance. We used the barrier parameters without external electrical field to calculate the tunneling resistance of electrons in the manuscript. It is not the value at Ohmic contact mode. According to the updated DFT calculation results, the CNT/MoS₂ heterojunction achieves Ohmic contact under the positive electric field of 0.2 V/Å. The corresponding barrier width w_t and barrier height Φ_t are 1.87 Å and 4.03 eV, respectively. We updated these parameters, and the calculated tunneling resistivity of electrons is $r_t = 2.882 \times 10^{-9} \Omega \cdot \text{cm}^2$.

Revisions in the manuscript:

1. The calculated results have been updated in the manuscript and the Supplementary Note 7.
2. The Electrostatic potential of the (5,5) CNT/MoS₂ heterostructure under applied electric field

of 0.2 V/\AA has been added as Supplementary Figure 10.

Supplementary Fig. 10 Electrostatic potential of the (5,5) CNT/MoS₂ heterostructure under applied electric field of 0.2 V/\AA . The potential barrier width w_t and barrier height Φ_t are 1.87 \AA and 4.03 eV , respectively.

Comment 5: The y-axis of Supplementary Fig. 10 may be mislabeled. Please double check.

Response 5: We have carefully checked Supplementary Fig. 10, and there is no mistake. The calculated transfer length L_T is indeed on the micron scale.

We thank Reviewer 3 again for his/her valuable comments, which have improved the quality of our manuscript.

Reviewers' Comments:

Reviewer #1:

Remarks to the Author:

The authors have resolved all issues I raised in the previous review. So, the manuscript is ready to be published.

Reviewer #2:

Remarks to the Author:

I thank the reviewer for the comprehensive revisions. I'm happy with the manuscript in its current form.

Reviewer #3:

Remarks to the Author:

In the revised manuscript, the authors have made tremendous additional effort to address the reviewers' comments. I really appreciate all the efforts that the authors have made. Most of the concerns raised by the reviewers have been resolved. The manuscript now is in a much better shape and are almost ready for publication. I just have one minor comment:

In the response 2 of reviewer 3, the authors mentioned that they have tried to fabricate devices with other contact metal and presented some results. The dimension of the devices (channel length) are too large, making the the overall performance less contributed by the contact, but rather by the channel resistance. Therefore the comparison of the CNT contact and the other contact metal are still a little misleading. If the authors cannot reproduce other people's work as the controlled group with similar performance, I would suggest the authors to remove their control group and just cite other people's results as comparison.

Response to Reviewer 3

Comment: In the revised manuscript, the authors have made tremendous additional effort to address the reviewers' comments. I really appreciate all the efforts that the authors have made. Most of the concerns raised by the reviewers have been resolved. The manuscript now is in a much better shape and are almost ready for publication. I just have one minor comment:

In the response 2 of reviewer 3, the authors mentioned that they have tried to fabricate devices with other contact metal and presented some results. The dimension of the devices (channel length) is too large, making the overall performance less contributed by the contact, but rather by the channel resistance. Therefore, the comparison of the CNT contact and the other contact metal are still a little misleading. If the authors cannot reproduce other people's work as the controlled group with similar performance, I would suggest the authors to remove their control group and just cite other people's results as comparison.

Response: We thank Reviewer 3 very much for his/her kind and valuable comments on our manuscript. As shown in Supplementary Fig. 19, the MoS₂ FETs with CNT contacts (<2nm in contact length) exhibit close on-state current and larger on/off ratio compared to the devices using Ni and Au contacts (3 μm in contact length). The transfer curves of CNT-MoS₂ FETs and metal-MoS₂ FETs was measured by electrodes 1-2 and A-B, respectively. It should be noted that the CNT-MoS₂ FET exhibits better switching performance than the Ni-MoS₂ FET, even with longer channel length and narrower channel width. This is sufficient to demonstrate that CNT contacts have lower contact resistance. Due to the different experimental conditions, our control experiments may not achieve the optimal value reported in the literature. Therefore, we have added references on MoS₂ FET with Ti and Ni contacts. The contact resistance (R_c) between Ni, Ti and MoS₂ has also been supplemented to the revised Figure 3e to provide a complete comparison.

Revisions in the manuscript:

1. The description of Supplementary Figure 19 has been added in the 1st paragraph of the **Electrical transport measurements** section. “**Supplementary Figure 19 also shows the better switching performance of CNT contacts by comparing the devices using CNT, Ni and Au contacts.**”
2. We have added the contact resistance data of Ti and Ni contacts in Figure 3e. References 21 and 52 have been added. Revisions have also been made to the last paragraph of the **LTM and extraction of contact resistance** section. “**Figure 3e compares the contact resistance R_c of this work with the state-of-the-art contacts in literatures including Ti, Ni, Au, In, Graphene, Bi and CNT bundles.**”